# A neural command circuit for grooming movement control

**Stefanie Hampel[1], Romain Franconville[1], Julie H Simpson[1,2]\*, Andrew M Seeds[1]\***

[1]Janelia Research Campus, Howard Hughes Medical Institute, Ashburn, United States; [2]Department of Molecular, Cellular, and Developmental Biology, University of California, Santa Barbara, Santa Barbara, United States

**Abstract** Animals perform many stereotyped movements, but how nervous systems are organized for controlling specific movements remains unclear. Here we use anatomical, optogenetic, behavioral, and physiological techniques to identify a circuit in *Drosophila melanogaster* that can elicit stereotyped leg movements that groom the antennae. Mechanosensory chordotonal neurons detect displacements of the antennae and excite three different classes of functionally connected interneurons, which include two classes of brain interneurons and different parallel descending neurons. This multilayered circuit is organized such that neurons within each layer are sufficient to specifically elicit antennal grooming. However, we find differences in the durations of antennal grooming elicited by neurons in the different layers, suggesting that the circuit is organized to both command antennal grooming and control its duration. As similar features underlie stimulus-induced movements in other animals, we infer the possibility of a common circuit organization for movement control that can be dissected in *Drosophila*.

**\*For correspondence:**
simpsonj@janelia.hhmi.org (JHS);
seeds.andrew@gmail.com (AMS)

**Competing interests:** The authors declare that no competing interests exist.

## Introduction

An animal may perform a particular movement in response to its environment and internal state, and many movements are selected from a repertoire of stereotyped motor patterns. This repertoire can include movements that serve particular purposes, such as feeding, grooming, song production, locomotion, and even coordinated facial poses for expressing different emotions (*Grillner and Wallén, 2004*; *Grillner et al., 2005*). Many of these movements are produced by neural networks called pattern generators, which control the precise timing of motor neuron activity to coordinate stereotyped patterns of muscle contractions (*Pearson, 1993*; *Kiehn and Kullander, 2004*). These neural networks are localized to distinct regions of the central nervous system (CNS), such as the hindbrain and spinal cord in vertebrates and ventral nerve cord in arthropods, and are capable of producing their respective movements even when experimentally isolated from the brain and sensory inputs (*Grillner et al., 2005*; *Marder et al., 2005*; *Büschges et al., 2011*). The activity of pattern generators can be initiated, or adapted to particular circumstances, in response to inputs from proprioceptive neurons, neuromodulators, or command-like neurons (*Marder et al., 2005*; *Grillner, 2006*; *Ritzmann and Büschges, 2007*; *Blitz and Nusbaum, 2011*; *Harris-Warrick, 2011*).

The proximal trigger of pattern generator activity arises from the command-like neurons, which can consist of individual neurons, or populations of neurons that induce or 'command' specific movements (*Kupfermann and Weiss, 1978*; *Jing, 2009*). Such neurons are also implicated in controlling different parameters of the movements that they induce, such as their speed or duration (*Kupfermann and Weiss, 1978*; *Pearson, 1993*; *Kristan, 2008*; *Jing, 2009*). However, the organizational principles underlying how such command circuitry can both initiate movements and control their parameters remain unclear. Moreover, in many cases in which the activity of command-like neurons has been

**eLife digest** Many movements that animals perform regularly—including walking and grooming—consist of stereotyped sequences of muscle contractions. For example, a dog may scratch its side in response to a fleabite or because it is itchy. But how does the nervous system trigger such specific movements from among the repertoire of different movements that the animal could perform? It also remains unclear how such movements can be produced in a reliable, yet flexible manner.

Hampel et al. have now described the neural circuit that triggers and controls the stereotyped leg movements that the fruit fly *Drosophila* uses to groom its antennae. Such grooming movements are stereotyped yet have certain degrees of flexibility, which makes them ideal to study the neural circuits that underlie specific movements. Grooming further lends itself to this kind of investigation because it can be triggered by irritating the surface of the fly's body. There is also an extensive genetic toolkit that can be used to manipulate and observe the fruit fly's nervous system in detail.

Hampel et al. first identified sensory neurons in the flies' antennae that were needed to elicit grooming in response to irritating displacements of the antennae. Once these neurons were found, techniques—including those that allow specific neurons in the fly's brain to be precisely controlled—were then used to find other neurons that participate in the grooming process. This approach highlighted three groups of interneurons: two in the brain and one in the fly's equivalent of the spinal cord. Together these layers of sensory neurons and interneurons formed a circuit that triggered grooming whenever the antennae were disturbed.

Notably, activating different sensory or interneurons triggered bouts of antennal grooming of differing durations. This shows that the same neural circuit can both produce highly specific movements and modify the movements to provide flexibility. Neural circuits with similar features have been observed previously to induce other animal behaviors, for example, swimming in leeches. This suggests that this organization may be common in circuits that elicit movements. Additional experiments are now needed to validate whether similar circuits underlie other stereotyped movements in fruit flies and other animals.

experimentally manipulated (*Jing, 2009*), the behavioral impact of the manipulations was not assessed in intact and freely moving animals.

The development of neurogenetic tools in *Drosophila* has led to rapid progress in identifying command-like neurons. This progress has been enabled by the use of optogenetic and thermogenetic activation of specific, genetically targeted populations of neurons in freely moving adult flies (*Flood et al., 2013a*; *Owald et al., 2015*). For example, activation of specific neuronal types can elicit stereotyped movements such as feeding, locomotion, courtship song, or escape (*Lima and Miesenböck, 2005*; *von Philipsborn et al., 2011*; *Flood et al., 2013b*; *Gao et al., 2013*; *Inagaki et al., 2014*; *Bidaye et al., 2014*; *von Reyn et al., 2014*). Although some of these studies have led to the identification of individual neurons and groups of neurons that command their respective behaviors, the anatomical organization of and functional connections among such groups of neurons that control these movements has been largely unexplored.

Grooming movements (a.k.a. cleaning, scratching, or wiping reflexes) can be studied to determine the neuronal mechanisms by which specific movements are initiated and controlled. Grooming is ubiquitous among limbed animals as a means of protecting the body surface from different types of mechanical or chemical irritants (*Sachs, 1988*). Such sensory stimuli induce the movement of a limb to the irritated body part, which then scratches or wipes the surface (*Stein, 1983*; *Dürr and Matheson, 2003*). Because grooming movements can be predictably elicited by defined stimuli, they offer a way to access the sensory-connected neural circuitry that commands precisely targeted limb movements. In addition, grooming movements exhibit differing response durations, limb trajectories, stimulus-induced habituation, and can be suppressed, suggesting that the neural circuitry underlying these movements is subject to regulation and flexible control (*Sherrington, 1906*; *Stein, 1983*; *Corfas and Dudai, 1989*; *Page and Matheson, 2009*; *Seeds et al., 2014*). Therefore, the study of grooming movements may reveal basic principles of movement control, but little is known about the neuronal mechanisms governing their initiation and modulation.

We previously discovered that activating different neuronal populations in the fly nervous system could induce distinct grooming movements, such as grooming of the eyes, antennae, wings, thorax, or legs (*Seeds et al., 2014*). This raised the possibility that the functional organization of the neural circuitry controlling specific grooming movements could be defined. Here we examine a circuit that commands one of these grooming movements—antennal grooming. This movement involves the grasping and brushing of the antennae with the legs in response to different types of irritants (*Robinson, 1996*; *Böröczky et al., 2013*; *Seeds et al., 2014*). To deconstruct the neural circuitry underlying antennal grooming, we isolated a small number of GAL4-expressing transgenic lines that could elicit the appropriate leg movements when driving expression of the temperature-gated neuronal activator dTrpA1 (*Hamada et al., 2008*; *Seeds et al., 2014*). However, these lines expressed GAL4 in multiple neuronal subsets, making it difficult to determine which neurons were responsible for the grooming movement. In this work, we refine these GAL4 lines to identify the specific neurons that elicit antennal grooming. We show that these neurons are functionally connected to form a circuit that detects displacement of the antennae via mechanosensory neurons and then commands grooming through three different interneuronal classes. Our analysis of the complex organization of this circuit provides insight into how stereotyped movements are controlled.

## Results

### A group of Johnston's Organ mechanosensory neurons elicits antennal grooming

Given that grooming is induced by stimulation of the body surface, we first sought to identify sensory neurons that could relay such stimulation from the antennae to the brain. To this end, we revisited the GAL4 lines we identified in our previous behavioral screen and examined them for expression in the antennae (*Seeds et al., 2014*). One line expressed GAL4 in mechanosensory chordotonal neurons of the Johnston's Organ (JO, see below), and elicited antennal grooming when the targeted neurons were thermally activated by dTrpA1 (*Figure 1—figure supplement 1A,B*). Because this line also expressed in interneurons in the CNS (*Figure 1—figure supplement 1C,D*), we identified four additional GAL4 lines that target these sensory neurons by visually screening an image database of GAL4 line expression patterns (*Jenett et al., 2012*). Each of these lines elicited antennal grooming when used to thermogenetically activate the targeted neurons, further implicating this population of JO neurons in the behavior (*Figure 1—figure supplement 1G–J*, behavior not shown).

While our results were consistent with these sensory neurons being responsible for eliciting antennal grooming, the expression of GAL4 in other neurons allowed the possibility that they were contributing to this behavior. To examine whether the JO neurons elicit antennal grooming, we used the intersectional Split GAL4 (spGAL4) approach in an effort to restrict GAL4 activity to these neurons (*Luan et al., 2006*; *Pfeiffer et al., 2010*). Specifically, the activation domain (AD) of GAL4 is expressed in the genomic enhancer-driven pattern of one identified GAL4 line, while the DNA binding domain (DBD) of GAL4 is expressed in the pattern of another line (see 'Materials and methods'). When both halves are co-expressed in the same cell, the activity of GAL4 is reconstituted. By co-expressing the AD and DBD in the patterns of different enhancer pairs, we observed reconstitution of GAL4 activity in the JO neurons (*Figure 1—figure supplement 2A–E'*). Five of these different AD/DBD pairs showed increased antennal grooming with thermogenetic activation (*Figure 1A*, pairs referred to as aJO-spGAL4-1 through aJO-spGAL4-5, *Supplementary file 1* shows enhancer pairs used).

Confocal imaging confirmed that each spGAL4 pair targeted neurons within the second antennal segment, as revealed by staining for the neuronal protein Elav (*Figure 1C,D*, *Figure 1—figure supplement 2A'–E'*) (*Kamikouchi et al., 2006*). The anatomy of these neurons identifies them as subsets of the approximately 500 chordotonal neurons within the JO, a mechanosensory structure that detects antennal movements. All five pairs were expressed in two distinct clusters of 40–50 neurons each in the dorsal and ventral regions of the JO, and we designate these clusters collectively as the antennal grooming JO (aJO) (*Figure 1D*, *Figure 1—figure supplement 2A'–E'*). Although four of the pairs still targeted GAL4 activity to neurons outside of the JO (*Figure 1—figure supplement 2B–E*), aJO-spGAL4-1 expressed almost exclusively in the aJO, providing strong evidence that these sensory neurons elicit antennal grooming (*Figure 1D,E*). To independently evaluate the necessity of

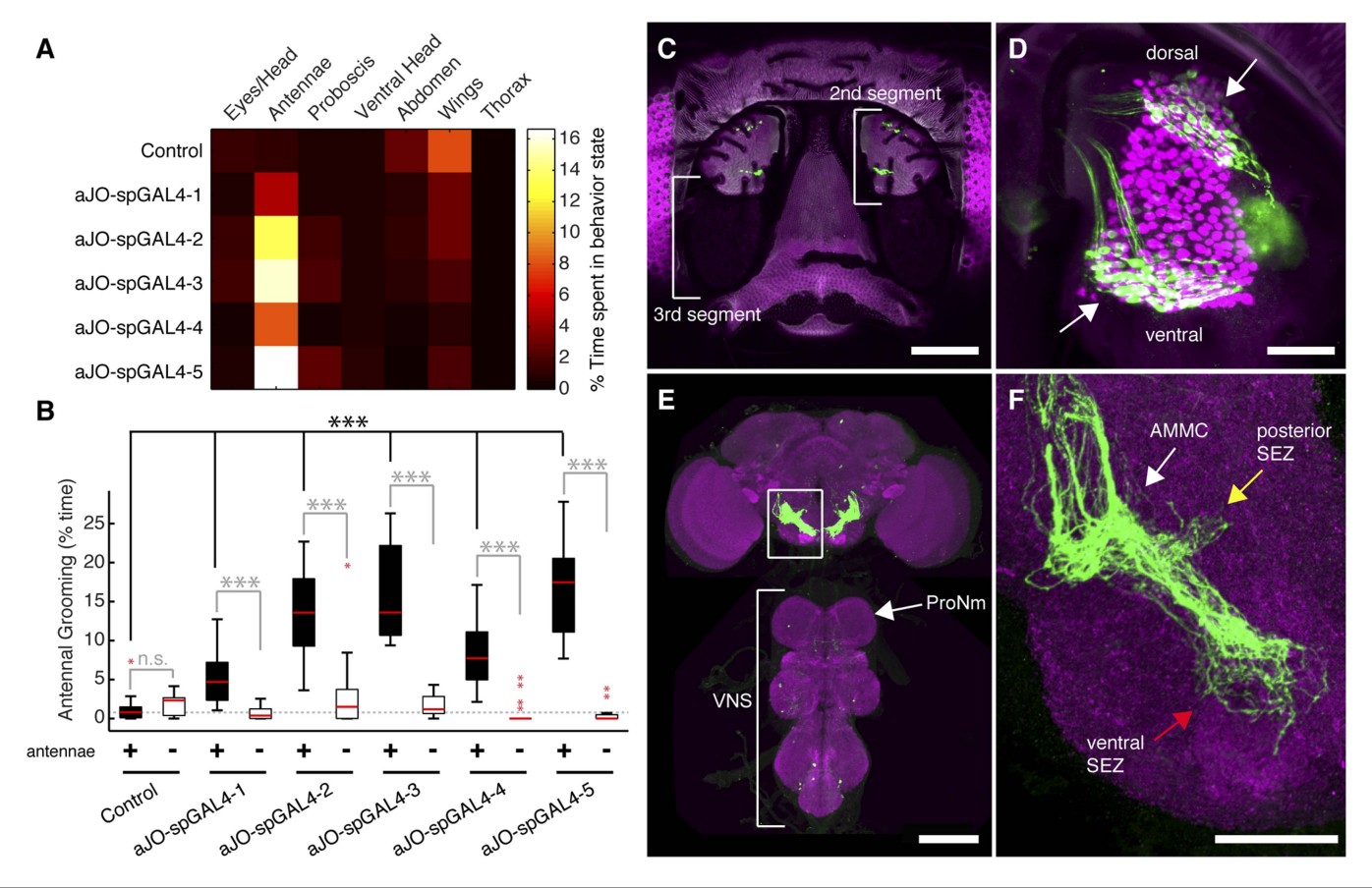

**Figure 1**. Sensory neurons that elicit antennal grooming. (**A**) Grooming movements performed by flies in which aJO spGAL4 pairs drove expression of thermally activated dTrpA1. Movements were manually scored from 2 min of recorded video per fly (n ≥ 17 flies per spGAL4). Colors correspond to the percent of total time spent performing each movement. (**B**) Percent time flies spent antennal grooming with thermogenetic activation of neurons targeted by spGAL4 pairs, with or without their antennae (filled or open boxes, respectively). Bottom and top of the boxes indicate the first and third quartiles respectively; median is the red line; whiskers show the upper and lower 1.5 IQR; red dots are data outliers (n ≥ 17 for each box; asterisks show p < 0.0001, Kruskal–Wallis and post hoc Mann–Whitney U pairwise tests with Bonferroni correction). Dotted line marks the median of the intact control. (**C–F**) aJO-spGAL4-1 driving expression of green fluorescent protein (GFP). Maximum intensity projections are shown. (**C**) Frontal view of the head (native GFP fluorescence, green; cuticle autofluorescence, magenta). Left bracket shows the third antennal segment. Right bracket marks the second antennal segment, which is shown in (**D**). Scale bar, 100 μm. (**D**) Second antennal segment co-stained with anti-GFP (green) and anti-Elav (magenta, marks neuronal nuclei) antibodies. White arrows show the ventral and dorsal aJO clusters. Scale bar, 25 μm. (**E, F**) Central nervous system (CNS) co-stained with anti-GFP (green) and anti-Bruchpilot (magenta) to visualize the aJO afferent projections into the ventral brain neuropile (**E**) and their specific targeting of the indicated antennal mechanosensory and motor center (AMMC) and subesophageal zone (SEZ) regions (arrows shown in **F**). Box in (**E**) indicates region shown in **F**. Scale bars, (**E**) 100 μm and (**F**) 25 μm. Prothoracic neuromeres (ProNm). Ventral nervous system (VNS). See also *Figure 1—figure supplement 1* and *Figure 1—figure supplement 2*.

The following figure supplements are available for figure 1:

**Figure supplement 1**. GAL4 lines that target expression to sensory neurons from the antennae and elicit grooming.

**Figure supplement 2**. spGAL4 pairs that target expression to sensory neurons in the antennae and elicit grooming.

**Figure supplement 3**. JO neurons projecting to zone C/E elicit antennal grooming.

**Figure supplement 4**. Most stochastically labeled aJO neurons show projections to both the AMMC and ventral SEZ.

the aJO in antennal grooming, we tested whether amputation of the antennae, and thus removal of the JO, would abolish the thermogenetically elicited behavior of the spGAL4 pairs. Indeed, antennal amputation abolished grooming at temperatures that would normally activate dTrpA1 and elicit

antennal grooming (*Figure 1B*). These experiments demonstrate that the aJO induces antennal grooming.

The JO comprises different subgroups of chordotonal neurons whose axons project to distinct zones (zones A–E) in the brain antennal mechanosensory and motor center (AMMC) (*Kamikouchi et al., 2006*; *Yorozu et al., 2009*; *Matsuo et al., 2014*). aJO axons enter the AMMC with zone C/E neurons (*Figure 1—figure supplement 3A–B*). However, unlike previously described C/E neurons that terminate in the AMMC, the aJO have three apparent projections within the ventral brain: the AMMC, the ventral subesophageal zone (SEZ), and the posterior SEZ (*Figure 1E,F*). To test whether each neuron within the aJO has all three projections, we performed multicolor stochastic labeling (*Nern et al., 2015*), which allows for visualization of individual neurons within the aJO population. Because each of the single neurons that we isolated projects from the AMMC to the ventral SEZ (*Figure 1—figure supplement 4A–F,I*), it would appear that the majority of neurons within the aJO have similar projections. A smaller subset appears to project from the AMMC to the posterior SEZ; however, we were unable to isolate individual cells to definitively show this (*Figure 1—figure supplement 4G,H*). In contrast to previously identified zone C/E-projecting neurons, we found no evidence of aJO neurons that project only to the AMMC. Because none of the previously described JO neurons project to the ventral SEZ (*Kamikouchi et al., 2006*; *Matsuo et al., 2014*), the aJO corresponds to a previously unrecognized set of neurons.

Given that aJO neurons project to zone C/E before passing to the SEZ, we tested whether activation of previously described C/E neurons could elicit antennal grooming. Indeed, activation of zone C/E neurons using published GAL4 drivers (*Kamikouchi et al., 2006*) elicited antennal grooming (*Figure 1—figure supplement 3C–E,H*). In contrast, activation of zone A or B neurons did not elicit antennal grooming (*Figure 1—figure supplement 3F,G,H*), indicating that only the zone C/E subpopulation elicits the behavior. Therefore, our data indicate that at least two types of zone C/E-projecting neurons are sufficient to induce grooming. The first terminates within the AMMC and constitutes a previously described set of JO neurons (*Kamikouchi et al., 2006*), whereas the second type are the aJO neurons that project into the AMMC and then ventrally to the SEZ.

## Three different interneuron classes elicit antennal grooming

The muscles that control front leg movements necessary for antennal grooming are innervated by neurons residing in the most anterior region of the ventral nervous system (VNS), the prothoracic neuromeres (ProNm) (*Figure 1E*) (*Burrows, 1996*; *Brierley et al., 2012*). Because JO afferent projections terminate in the brain and do not project to the ProNm, where they would be positioned to activate leg movements, we reasoned that additional neurons must project to the ProNm to command grooming behavior. Therefore, we sought to identify interneurons that transmit the sensory signal to the ProNm.

Of the GAL4 lines that elicited antennal grooming in our previous screen (*Seeds et al., 2014*), two lack expression in antennal sensory neurons but have expression in interneurons within the brain (lines R26B12 and R18C11, behavioral analysis in *Figure 1—figure supplement 1A*, expression patterns in *Figure 2—figure supplement 1A–C*). A visual screen of the GAL4 line expression pattern database (*Jenett et al., 2012*) identified additional lines with interneuron projection patterns in the AMMC and SEZ that made them candidates for associating with the JO projections. Five of these lines elicited antennal grooming when thermogenetically activated (*Figure 2—figure supplement 1A*). As each GAL4 line expressed in other neuronal populations (*Figure 2—figure supplement 1D–H*), we again generated spGAL4 lines using their respective enhancers to further restrict GAL4 activity to the behaviorally relevant interneurons. With this approach, we identified spGAL4 pairs that elicited antennal grooming with thermogenetic activation, and targeted interneurons that we designated aBN1, aBN2, and antennal descending neuron (aDN) for reasons described below (*Figure 2A*, *Supplementary file 1*). Of note, there were striking differences in the amounts of grooming elicited by these pairs (*Figure 2A*). For example, aDN1-spGAL4-1 flies spent 4.8% of their time grooming their antennae vs 66.1% for aBN2-spGAL4-2 (*Figure 2B*, black boxes).

To validate that the spGAL4 lines target interneurons, we removed sensory neurons by amputating the antennae, and then measured the time spent grooming during thermogenetic activation. As anticipated, amputation did not abolish movements directed towards the antennal region, and some pairs significantly increased such movements relative to intact antennae (*Figure 2B*,

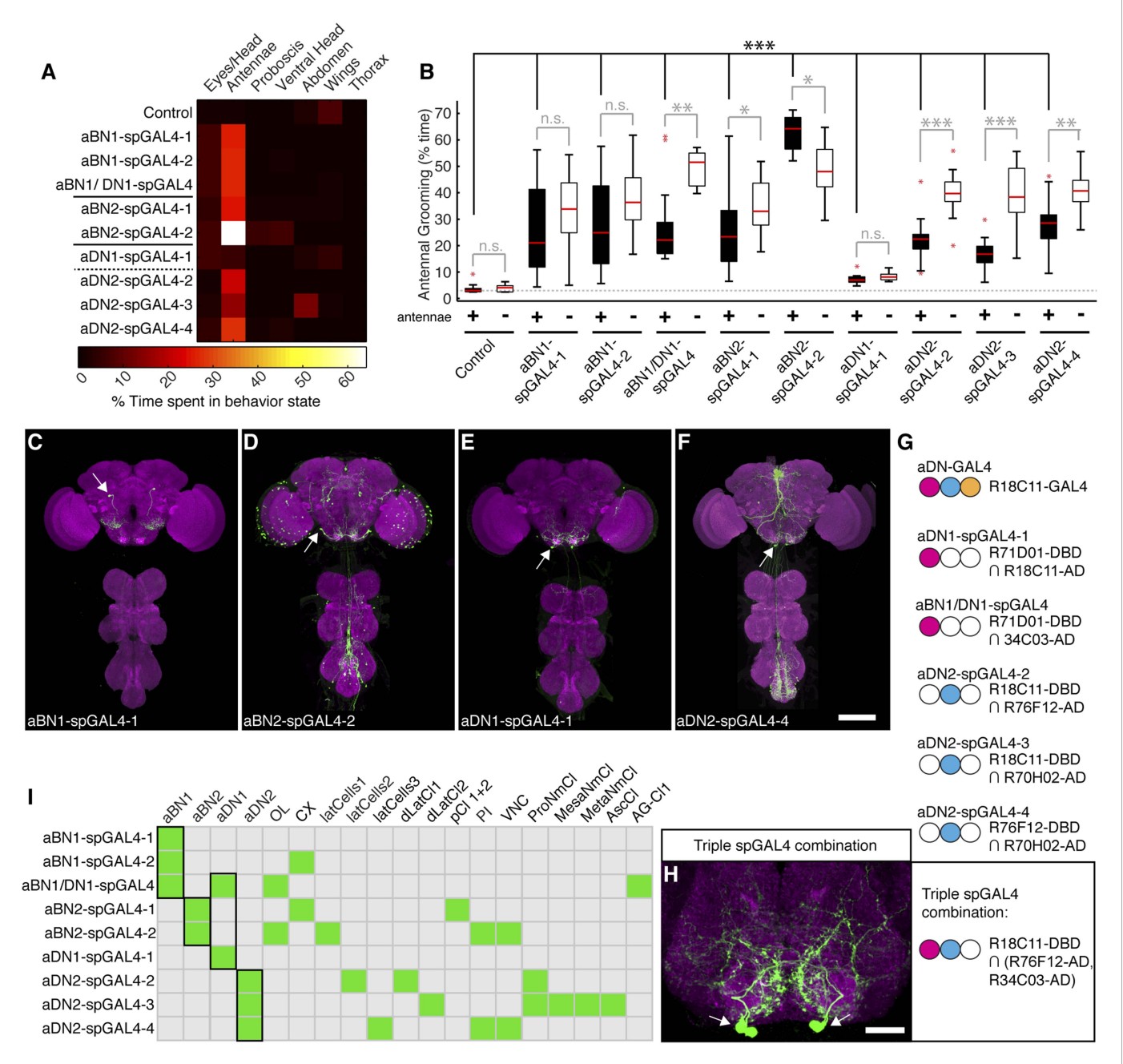

**Figure 2**. Interneurons that elicit antennal grooming. (**A**) Grooming movements performed by interneuron spGAL4 pairs expressing thermally activated dTrpA1. Data was obtained and displayed as described in *Figure 1A*. (**B**) Percent time flies spent antennal grooming with thermogenetic activation of interneurons targeted by spGAL4 pairs, with or without their antennae (filled or open boxes respectively). Box plots and statistics are described in *Figure 1B*. Asterisks represent the following p values: *p < 0.01, **p < 0.001, ***p < 0.0001 (n ≥ 9 flies per spGAL4). Black p value statistics show differences between control and spGAL4 flies with their antennae. Gray statistics show differences between each spGAL4 with and without the antennae. (**C**–**F**) GFP expression patterns of spGAL4 lines: (**C**) aBN1-spGAL4-1, (**D**) aBN2-spGAL4-2, (**E**) aDN1-spGAL4-1, (**F**) aDN2-spGAL4-4. Images show maximum intensity projections of co-staining with anti-GFP (green) and anti-Bruchpilot (magenta). White arrows show cell bodies. Scale bars, 100 μm. (**G**) antennal descending neurons (aDNs) targeted by each spGAL4 pair. Circles represent one of three neurons in aDN-GAL4. Filled circles show which neurons are targeted by each spGAL4 (enhancer pairs listed). (**H**) Two aDN neurons are targeted in a triple spGAL4 combination expressing GFP (white arrows). No spGAL4 combinations were identified that exclusively target aDN3. Scale bar, 25 μm. (**I**) Graphical summary of neuronal expression patterns of the spGAL4 pairs. Green boxes indicate expression of the pair on the left (rows) in the indicated neurons or region listed above the grid (columns). *Figure 2—figure supplement 2* shows the locations of these neurons. Black framing highlights antennal grooming neurons.

*Figure 2. continued on next page*

*Figure 2. Continued*

The following figure supplements are available for figure 2:

**Figure supplement 1**. GAL4 lines that elicit antennal grooming.

**Figure supplement 2**. spGAL4 lines with interneuron expression that elicit antennal grooming.

aBN1/aDN1-spGAL4, aBN2-spGAL4-1, and aDN2-spGAL4-2/3/4 show increased grooming). Thus, we conclude that these spGAL4 pairs target interneurons that induce antennal grooming when activated. Moreover, the increased grooming associated with antennal amputation raises the possibility that grooming might be negatively influenced by sensory feedback from the antennae.

We next examined the anatomy of these interneurons in greater detail. Two classes that induce antennal grooming are located entirely in the brain, and we designate them antennal grooming brain interneurons 1 and 2 (aBN1 and aBN2). aBN1-spGAL4-1 targeted expression to aBN1, a single interneuron in the ventral brain (*Figure 2C*). Two other spGAL4 pairs also targeted aBN1 and were able to elicit antennal grooming with thermogenetic activation (aBN1-spGAL4-2, aBN1/aDN1-spGAL4, *Figure 2I*, *Figure 2—figure supplement 2A–C*). aBN2 was targeted by two spGAL4 pairs that each used the R26B12 enhancer, which itself expresses in a cluster of eight neurons with cell bodies in the posterior and ventrolateral brain. Each pair targets expression to either three or five of these neurons (aBN2-spGAL4-1, aBN2-spGAL4-2, *Figure 2D,I*, *Figure 2—figure supplement 2D,E*, *Figure 4—figure supplement 2C,D*).

The other interneuron class comprises aDNs that project from the brain to the VNS. Five spGAL4 pairs targeted single descending neurons with cell bodies located in the posteroventral SEZ (*Figure 2E,F,I*, *Figure 2—figure supplement 2C,F–I*). Given that the R18C11 enhancer was used to generate several of these pairs, and by itself targets expression to three aDNs when driving GAL4 (*Figure 2—figure supplement 1C*), we asked whether each spGAL4 pair targeted the same neuron or distinct neurons. By simultaneously combining the GAL4 DBD, expressed under control of the R18C11 enhancer, with two versions of the GAL4 AD, one expressed under control of the R76F12 enhancer and the other controlled by the R34C03 enhancer, we observed two aDNs (*Figure 2G,H*). This demonstrates that the spGAL4 pairs target distinct descending interneurons, and we named these aDN1 and aDN2. However, we did not identify a spGAL4 combination that exclusively expresses in the third R18C11-targeted aDN (named aDN3).

## A putative circuit that elicits grooming in response to antennal displacement

The identified neuronal classes all have projections in the ventral brain (*Figure 3A–D*). aBNs project to the AMMC and SEZ, following the aJO projections (*Figure 3B,C*), whereas two aDNs (aDN1 and aDN2), and likely aDN3 project both to the SEZ and through the cervical connective to the ProNm (*Figure 3D*, only aDN1 shown). Manual and computational alignment of the projections from these classes suggested intimate associations and the potential to form a neural circuit (*Figure 3E*, *Video 1*, *Video 2*). Thus, we explored their functional relationships.

We first asked whether the grooming elicited by aJO thermogenetic activation required the activity of the aBNs and aDNs by performing a behavioral epistasis test in which we genetically silenced the activity of the interneurons while activating the aJO (*Figure 4A*). For silencing the interneurons, we used the aBN spGAL4 pairs and aDN-GAL4 to target expression of the synaptic transmission blocker, tetanus toxin (TNT) (*Sweeney et al., 1995*). To genetically access the aJO independent of the interneurons, we employed the LexA binary transcriptional system so that the aJO enhancer R27H08 directed expression of LexA (*Lai and Lee, 2006*; *Pfeiffer et al., 2010*). The resulting aJO-LexA line targeted aJO neurons and also elicited antennal grooming with thermogenetic activation (*Figure 4—figure supplement 1A,B*). Although aJO-LexA targets more JO neurons than the aJO spGAL4 pairs (*Figure 4—figure supplement 2A,B*), expression of TNT in the aJO by aJO-spGAL4-1 significantly decreased antennal grooming in response to thermogenetic activation by aJO-LexA. This showed that the aJO-spGAL4-1 neurons constituted a major portion of the grooming elicited with aJO-LexA (*Figure 4B*, blue boxes). However, given that aJO-LexA targets additional zone C/E

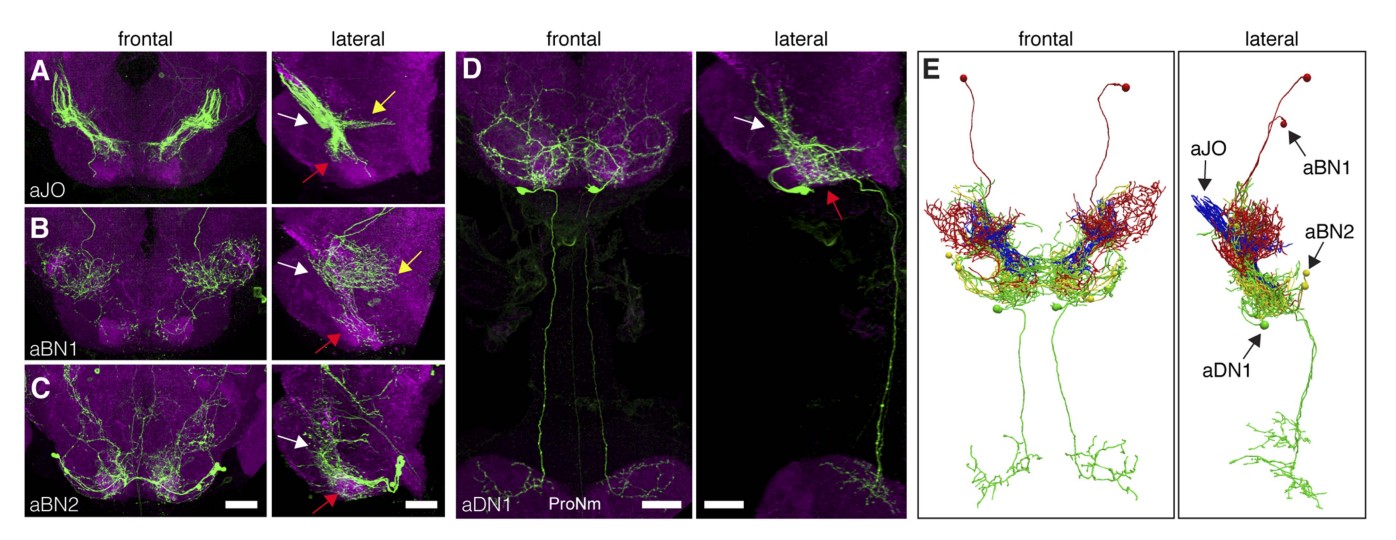

**Figure 3**. Neurons that elicit antennal grooming have neurites in the AMMC and/or SEZ. (**A–D**) spGAL4 pairs targeting each neuronal class in the ventral brain: (**A**) aJO-spGAL4-1, (**B**) aBN1-spGAL4-1, (**C**) aBN2-spGAL4-2, and (**D**) aDN1-spGAL4-1. aDN1 is shown as an example in (**D**), but there are additional aDNs (aDN2 and aDN3, see *Figure 2F* and *Figure 2—figure supplement 2G–I*). CNSs stained with anti-GFP (green) and anti-bruchpilot (magenta). Maximum intensity projections are shown from frontal and lateral views. Arrows show the different projection regions: AMMC (white), posterior SEZ (yellow), and ventral SEZ (red). Scale bars, 25 µm. (**E**) Traced neurons in different colors manually aligned (also shown in *Video 1*).

neurons that could induce grooming (*Figure 1—figure supplement 3C–E,H*), it is possible that they are responsible for the residual grooming that occurs with expression of TNT in aJO-spGAL4-1 (*Figure 4B*, blue boxes). Therefore, we hereafter refer to the neurons targeted by aJO-LexA as aJO+ C/E neurons. We next found that TNT expression in the interneurons aBN1 and aBN2 suppressed grooming when aJO+C/E neurons were thermogenetically activated (*Figure 4B*, red and yellow, respectively). However, grooming was not suppressed when TNT was expressed in the aDNs (*Figure 4B*, green boxes). We conclude that the aBNs are necessary for the grooming response to aJO+C/E activation, while additional descending neurons important for mediating antennal grooming may remain to be identified.

We next sought to identify a sensory stimulus upstream of this putative grooming circuit. Because JO neurons detect antennal movements (*Kamikouchi et al., 2009*; *Yorozu et al., 2009*; *Matsuo et al., 2014*), we reasoned that the role of the aJO might be to elicit grooming in response to physical displacements of the antennae. To test this, we glued iron powder to the third antennal segments and tethered the flies within an electromagnetic behavioral set up (*Figure 4C,D*). Application of a magnetic field at 1 Hz caused visible displacements of the antennae (31 ± 6 µm) and elicited grooming of both the antennae and other head parts (*Figure 4E*, white boxes, *Figure 4—figure supplement 3C,D*, *Video 3*). The aJO was critical for this response because expression of TNT in these neurons significantly reduced antennal and head grooming in response to the magnetic field (*Figure 4E*, blue boxes, head grooming not shown). Given that activation of aJO elicits antennal grooming almost exclusively (*Figure 1A*, *Figure 7—figure supplement 1A,B*), the role of the aJO in head grooming is unclear.

We next examined the functional necessity of the different interneuron classes for grooming in response to antennal displacement. aBN1 was found to be necessary, as expression of TNT in aBN1-spGAL4-1 or aBN1-spGAL4-2 both reduced the grooming response (*Figure 4E*, red boxes). Intriguingly, expression of TNT targeted by the two different aBN2 spGAL4 pairs gave opposing results in this assay (*Figure 4E*, yellow boxes): aBN2-spGAL4-1 activity showed necessity for antennal grooming while aBN2-spGAL4-2 did not. This contrasted with our earlier finding that both spGAL4 pairs disrupted grooming when the aJO+C/E neurons were thermogenetically activated (*Figure 4B*, yellow boxes). This may reflect a functional difference between the specific subsets of aBN2 neurons

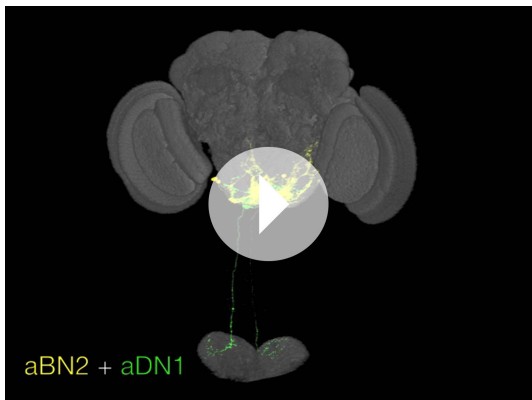

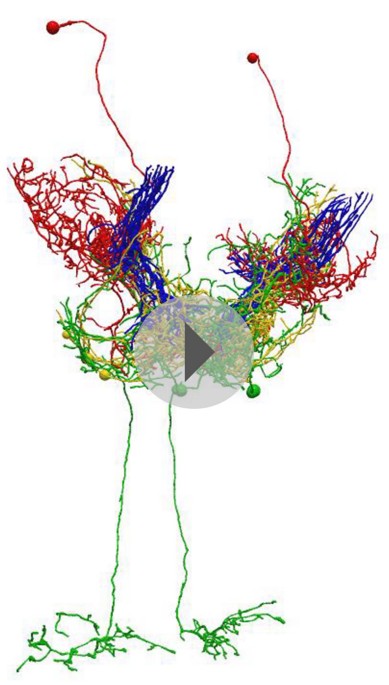

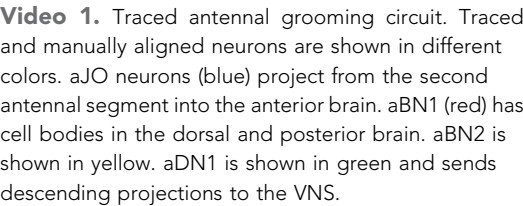

**Video 2.** Computationally aligned antennal grooming circuit. Computationally aligned neurons are shown in different colors. aJO neurons (blue), aBN1 (red), aBN2 (yellow) and aDN1 (green). The neuropil was stained with anti-bruchpilot (grey). See 'Materials and methods' for a description of how the computational alignment and rendering of images were done.

**Video 1.** Traced antennal grooming circuit. Traced and manually aligned neurons are shown in different colors. aJO neurons (blue) project from the second antennal segment into the anterior brain. aBN1 (red) has cell bodies in the dorsal and posterior brain. aBN2 is shown in yellow. aDN1 is shown in green and sends descending projections to the VNS.

targeted by these two spGAL4 pairs in the context of antennal displacement. Similar to the results for thermogenetic activation of aJO+C/E neurons, expression of TNT (or the inward rectifying potassium channel Kir) in all three aDNs did not block the grooming response to antennal displacement (*Figure 4E*, green box, Kir data not shown). Taken together, our results demonstrate that all identified neurons within the putative circuit elicit antennal grooming when activated, but not all are necessary for the grooming response to antennal displacement.

## Functional connectivity among the antennal grooming neuronal classes

To address the potential for connectivity among these neurons, we first examined the relative proximities of their projections in the brain by visualizing expression of LexA in one neuronal class and spGAL4 in another. We found that the aBN1 co-localized with all major aJO projections, whereas aBN2 co-localized with the aJO, AMMC, and ventral SEZ projections (*Figure 5A,B*, arrows). Interestingly, co-visualization of aJO with aDN1 or aDN2 revealed that the aDNs have distinct projections: aDN1 co-localizes with the aJO in the AMMC and ventral SEZ, whereas aDN2 only associated with the most ventral SEZ projections (*Figure 5C,D*, arrows). We confirmed that aDN1 projects more dorsally than aDN2 by examining their relative projections within the aDN-LexA pattern (*Figure 4—figure supplement 2E,F*). In support of our conclusion that the aBNs and aDNs are in close proximity with the JO projections, GFP-positive staining indicates reconstitution across synaptic partner experiments reported membrane contact among these neurons (*Figure 5—figure supplement 1A–D*) (*Feinberg et al., 2008*; *Gordon and Scott, 2009*).

We next examined projections of the interneuron types. For this, lines targeting LexA to either aBN2 or aDN were generated using the appropriate enhancers (*Supplementary file 1*). Both of these lines elicit antennal grooming with thermogenetic activation, and aDN-LexA targets three aDN neurons, while aBN2-LexA targets a cluster of four aBN2 neurons (*Figure 4—figure supplement 1A,C,D*, *Figure 4—figure supplement 2C–F*). Projections of aBN1 and aBN2 were closely associated in both the

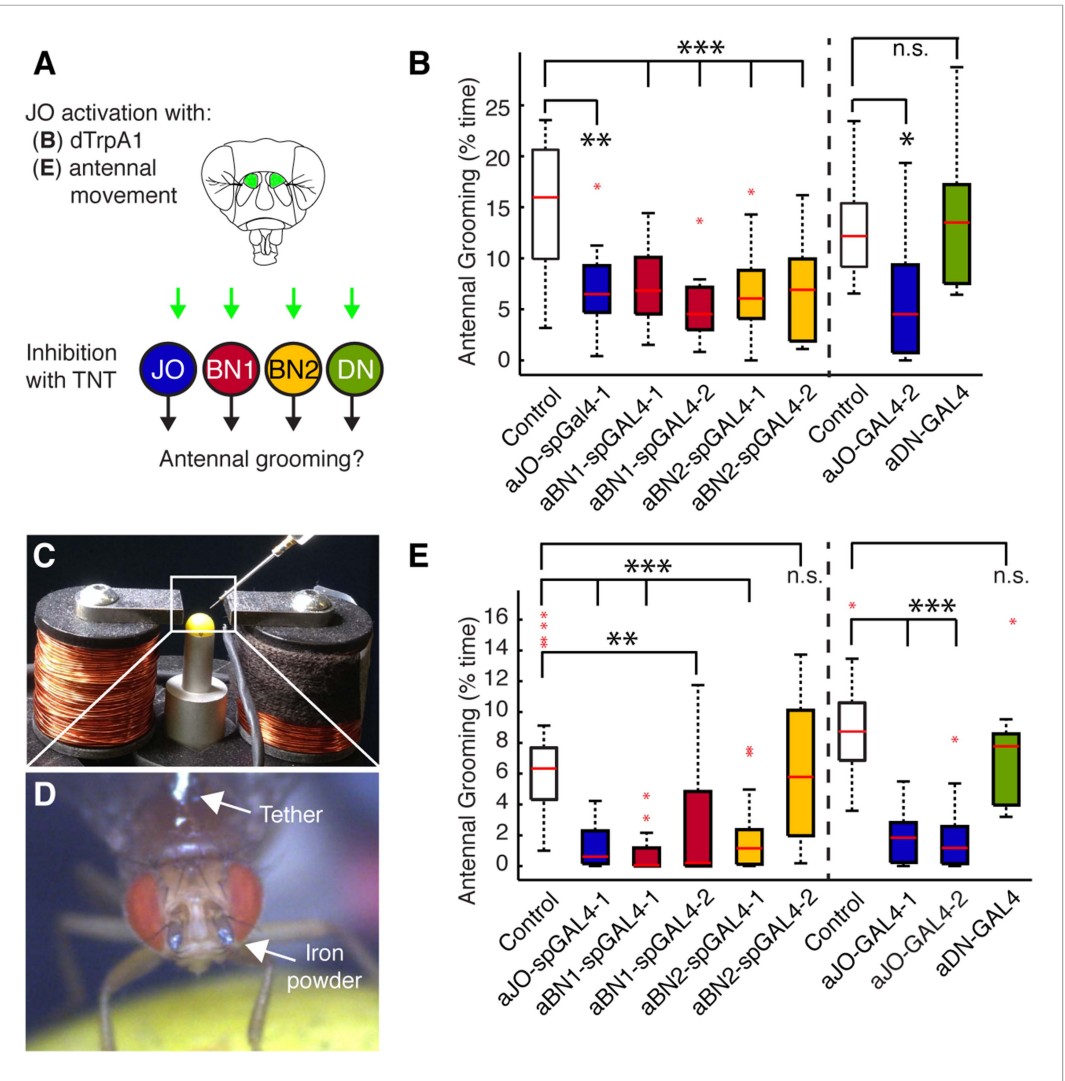

**Figure 4**. Functional relationships among putative antennal circuit components. (**A**) Overview of experiments shown in (**B**, **E**). Grooming was induced by thermogenetic activation of Johnston's Organ (JO) neurons (dTrpA1) or by imposed displacements of the antennae. Synaptic release was blocked in different neuronal classes expressing tetanus toxin (TNT). (**B**) Antennal grooming performed by flies with thermogenetic activation of the aJO while inhibiting synaptic release in interneuron classes. The experiment was performed and data is displayed as described in **Figure 1B** (n ≥ 11 flies per spGAL4). (**C–E**) To displace the antennae, iron powder was glued to the third antennal segment and the flies were tethered within an electromagnet. (**C**) Image of the electromagnet apparatus. (**D**) Tethered fly with iron powder on its antennae. Magnetic pulses were delivered to displace the third antennal segment at 1 Hz for 4 × 10 s periods, with 30 s rests between stimulations. Flies were recorded and their grooming movements were manually scored (see **Figure 4—figure supplement 3D** for ethograms). (**E**) The percent time that flies spent grooming their antennae within the total assay time is shown. The grooming responses to antennal movements were also tested while blocking synaptic release in the different neuronal types with TNT. Box plots and statistics are shown as described in **Figure 1B** (n ≥ 11 flies per line).

The following figure supplements are available for figure 4:

**Figure supplement 1**. aJO-, aBN2-, and aDN-LexA lines.

**Figure supplement 2**. Co-expression of LexA lines with selected spGAL4 pairs.

**Figure supplement 3**. Testing of stimulus parameters for the antennal displacement assay.

eLIFE Research article

Neuroscience

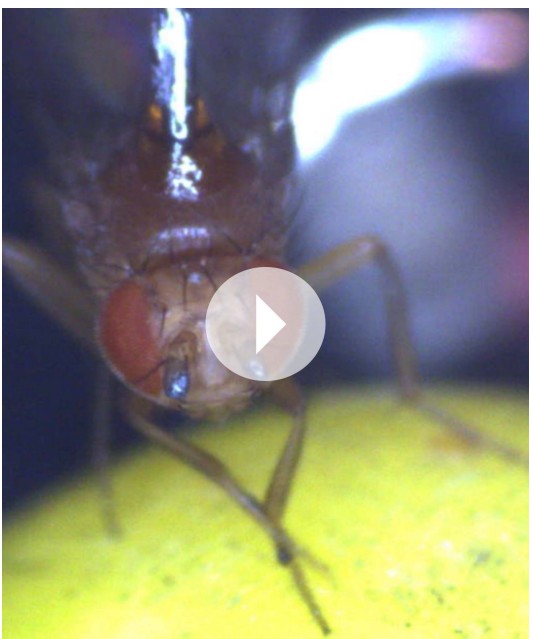

**Video 3.** Grooming movements performed in response to displacements of the antennae. The third antennal segments of a control fly were coated with iron powder, and the fly was tethered within the electromagnetic apparatus shown in *Figure 4C,D*. The infrared light positioned behind the fly shows when the magnetic field was applied to displace the antennae.

AMMC and ventral SEZ but lacked associations in the posterior SEZ (*Figure 5E*, arrows). Both aBN classes projected near the aDNs in the SEZ, but aDN1 also arborized near the dorsal projections of aBN2 (*Figure 5F–H*, arrows). We conclude that the projections of these sensory neurons and interneurons are in close proximity with each other, and potentially form a functionally connected circuit.

To test for functional connectivity among the neuronal classes, we measured calcium responses of the different interneurons when other neurons in the putative circuit were optogenetically activated. In isolated CNSs, we activated aJO+C/E neurons expressing the red light-inducible neuronal activator CsChrimson, while measuring fluorescence of the calcium responder GCaMP6s in the interneurons (*Chen et al., 2013*; *Klapoetke et al., 2014*). We detected significant calcium responses in aBN1, aBN2, and aDN1 (*Figure 6A–C*, *Figure 6—figure supplement 1A–D*, *Figure 6—figure supplement 2*, *Figure 6—figure supplement 3*), but only a weak response in aDN2 even with high-intensity red light (*Figure 6D*). Thus, aBN1, aBN2, aDN1 are likely downstream of aJO+C/E neurons, while aDN2 may be weakly or indirectly downstream of these sensory neurons.

We next tested whether CsChrimson-mediated optogenetic activation of either aBN1 or aBN2 induced a calcium response in the other. Activation of aBN1 induced a calcium response in aBN2 (*Figure 6E*), whereas only high-intensity red light activation of aBN2 could cause a weak calcium response in aBN1 (*Figure 6F*). The latter response was inconsistent, as only three out of five flies tested showed increases in calcium in aBN1 (average trace of three trials are shown in *Figure 6F*, raw traces of all five trials shown in *Figure 6—figure supplement 3*). Further, because the aBN2-LexA driver used to express CsChrimson in aBN2 also expresses in neurons in other parts of the brain (*Figure 4—figure supplement 1C*), we cannot rule out these other neurons as causing the aBN1 calcium response. Therefore, our results strongly support excitation from aBN1 to aBN2, but we only find weak functional imaging-based evidence supporting the reverse connection. Nevertheless, the excitatory responses that we observed are cholinergic because they were suppressed by the cholinergic receptor antagonist mecamylamine (*Figure 6E,F*).

We next tested whether aBNs are upstream of the aDNs. Activation of aBN1 induced a calcium response in the aDNs, and this excitatory response is cholinergic as it was suppressed by mecamylamine (*Figure 6G*). Interestingly, activation of aBN2 caused a mecamylamine-sensitive excitatory response in aDN1 (*Figure 6H*), yet decreased the basal fluorescence of GCaMP6s in aDN2 in a manner that was alleviated by mecamylamine (*Figure 6I*). The latter observation leads us to propose that aBN2 excites an unidentified neuron that then inhibits aDN2. We tested this possibility by applying picrotoxin, a GABA and inhibitory glutamate receptor blocker (*Cleland, 1996*; *Liu and Wilson, 2013*). In the presence of picrotoxin, activation of aBN2 abolished the decreased calcium responses in aDN2 (*Figure 6I*). This corroborated the presence of an inhibitory neuron (IN) that impinges on aDN2 that is stimulated by the activity of aBN2. Thus, the aDNs are subject to both feedforward excitation and indirect inhibition from aBN2. Taken together, these data demonstrate functional connectivity between the JO, aBNs, and aDNs and imply that they form a circuit that relays antennal sensory information through the brain and conveys it to the VNS.

Hampel *et al.* eLife 2015;4:e08758. DOI: 10.7554/eLife.08758

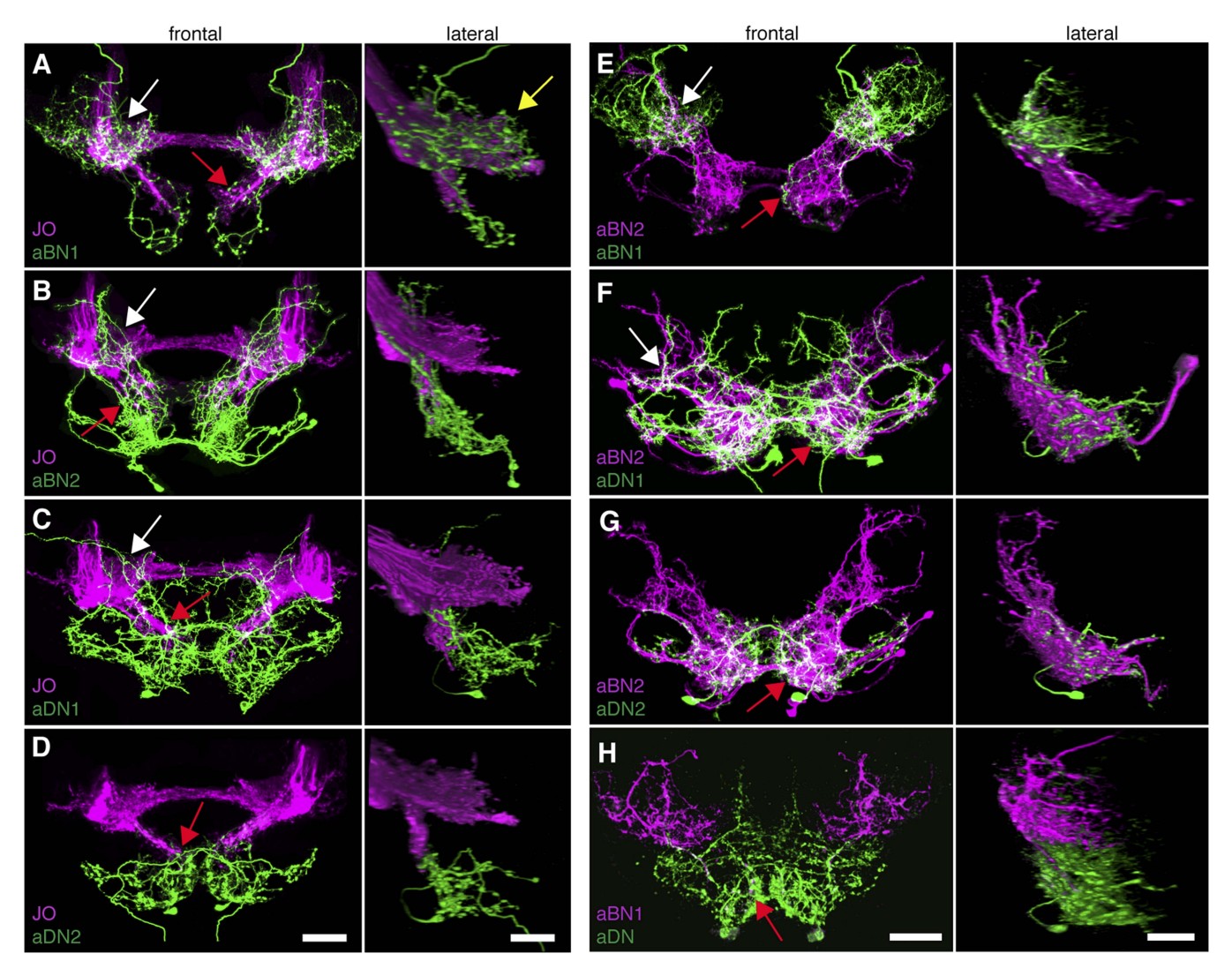

**Figure 5**. Antennal grooming neurons are in close proximity. (**A–H**) Co-expression in neuronal pairs using two binary expression systems (LexA and spGAL4) to express tdTomato or GFP in each neuronal class. Processed maximum intensity projections of frontal and lateral views are shown. See 'Materials and methods' about how images were processed (unprocessed images in *Figure 5—figure supplement 2*). Scale bars, 25 µm. (**A–D**) Proximity between aJO-LexA targeted sensory projections (magenta) and the following interneuron spGAL4 pairs (green): (**A**) aBN1-spGAL4-1, (**B**) aBN2-spGAL4-1, (**C**) aDN1-spGAL4-1 and (**D**) aDN2-spGAL4-2. (**E–G**) Proximity between aBN2-LexA targeted neurons (magenta) and the following interneuron spGAL4 pairs (green): (**E**) aBN1-spGAL4-1, (**F**) aDN1-spGAL4-1, (**G**) aDN2-spGAL4-2. (**H**) Proximity between aDN-LexA targeted neurons aDN (green) and aBN1-spGAL4-1 targeted neurons (magenta). Overlap between different projections of the LexA and spGAL4-targeted neurons is indicated by different colored arrows: (**A–C**, **E**, **F**) AMMC projections white arrows, (**A**) posterior SEZ projection (yellow arrow), (**B–H**) ventral SEZ projections (red arrows).

The following figure supplements are available for figure 5:

**Figure supplement 1**. GFP-positive staining indicates reconstitution across synaptic partner (GRASP) staining indicates close proximity of neurons involved in antennal grooming.

**Figure supplement 2**. Co-staining indicates close proximity of neurons involved in antennal grooming.

## A circuit whose components elicit different durations of antennal grooming

Our results motivated a circuit model that depicts the functional connectivity among the JO neurons, aBNs, and aDNs (*Figure 7A,B*). The model includes putative *reciprocal excitation* between the aBNs

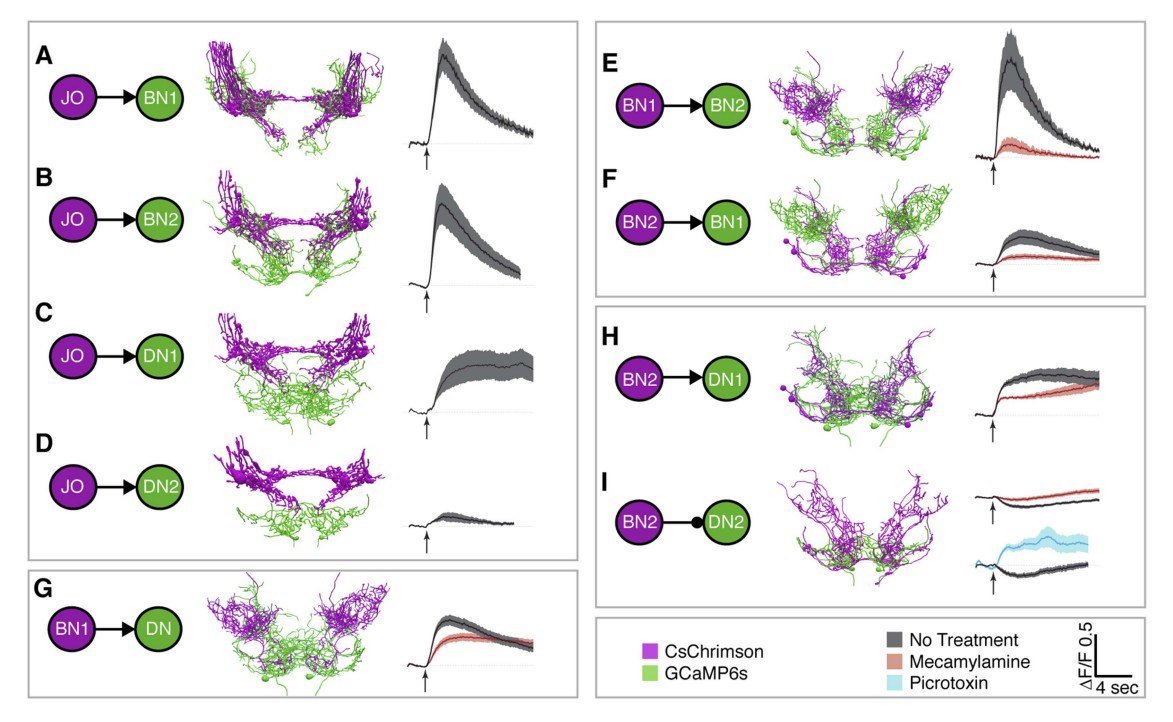

**Figure 6**. Different antennal grooming neurons are functionally connected. (**A–I**) Dissected CNSs with different neuronal classes expressing CsChrimson (magenta) were activated with red light while changes in calcium in their putative downstream partners expressing GCaMP6 (green) were imaged (ΔF/F). Each tested neuronal pair is shown using circles and as traced pairs. The direction of the connection and whether it is excitatory or inhibitory is depicted with an arrow (excitatory) or ball and stick (inhibitory). Changes in fluorescence of GCaMP6s of multiple flies under similar stimulus conditions are shown on the right (average ± s.e.m., 3–5 flies tested with 9–21 trials per trace). Arrow below each trace shows when the red light pulse was delivered. Black traces show flies that were imaged without drug treatment, whereas orange and blue traces were imaged while the nervous system was bathed with mecamylamine or picrotoxin respectively. See 'Materials and methods', *Figure 6—figure supplement 1*, *Figure 6—figure supplement 2*, *Figure 6—figure supplement 3*, and *Supplementary file 2* for detailed 'Materials and methods', stimulus conditions, and controls. (**A–D**) aJO-LexA tested with the following interneuron spGAL4 pairs: (**A**) aBN1-spGAL4-1, (**B**) aBN2-spGAL4-1, (**C**) aDN1-spGAL4-1, and (**D**) aDN2-spGAL4-2. (**E**, **F**) aBN2-LexA tested with aBN1-spGAL4-1. (**G**) aBN1-spGAL4-1 tested with aDN-LexA. (**H**, **I**) aBN2-LexA tested with either (**H**) aDN1-spGAL4-1 or (**I**) aDN2-spGAL4-2.

The following figure supplements are available for figure 6:

**Figure supplement 1**. Functional connectivity: controls, technical details, and raw data.

**Figure supplement 2**. Raw data for functional connectivity experiments (at low intensity red light).

**Figure supplement 3**. Raw data for functional connectivity experiments (at high intensity red light).

(*Figure 5E*, *Figure 6E,F*) and *feedforward inhibition* of aDN2 mediated by aBN2 and an unknown IN (*Figure 6I*). The proposed outputs of this circuit are the *parallel descending commands* to the VNS (*Figure 2*), where the antennal grooming pattern-generating circuitry is expected to reside (see 'Discussion'). We propose that the aDNs can act in parallel because experiments to thermogenetically activate aDN1 or aDN2 alone indicate that they are each sufficient to induce antennal grooming (*Figure 2B*). The presence of these features within the circuit suggests complex processing and raises the question of whether they influence the grooming output. Given that aBNs and aDNs induce different amounts of grooming with thermogenetic activation (*Figure 2A,B*), we postulated that one feature of the circuit is to control the duration of antennal grooming.

Our results raise the possibility that aBN1 and aBN2 are reciprocally excitatory, although the aBN2 to aBN1 excitatory connection was only weakly supported. One prediction of reciprocal excitation between the aBNs is that activation of either neuronal class will induce grooming that persists in the

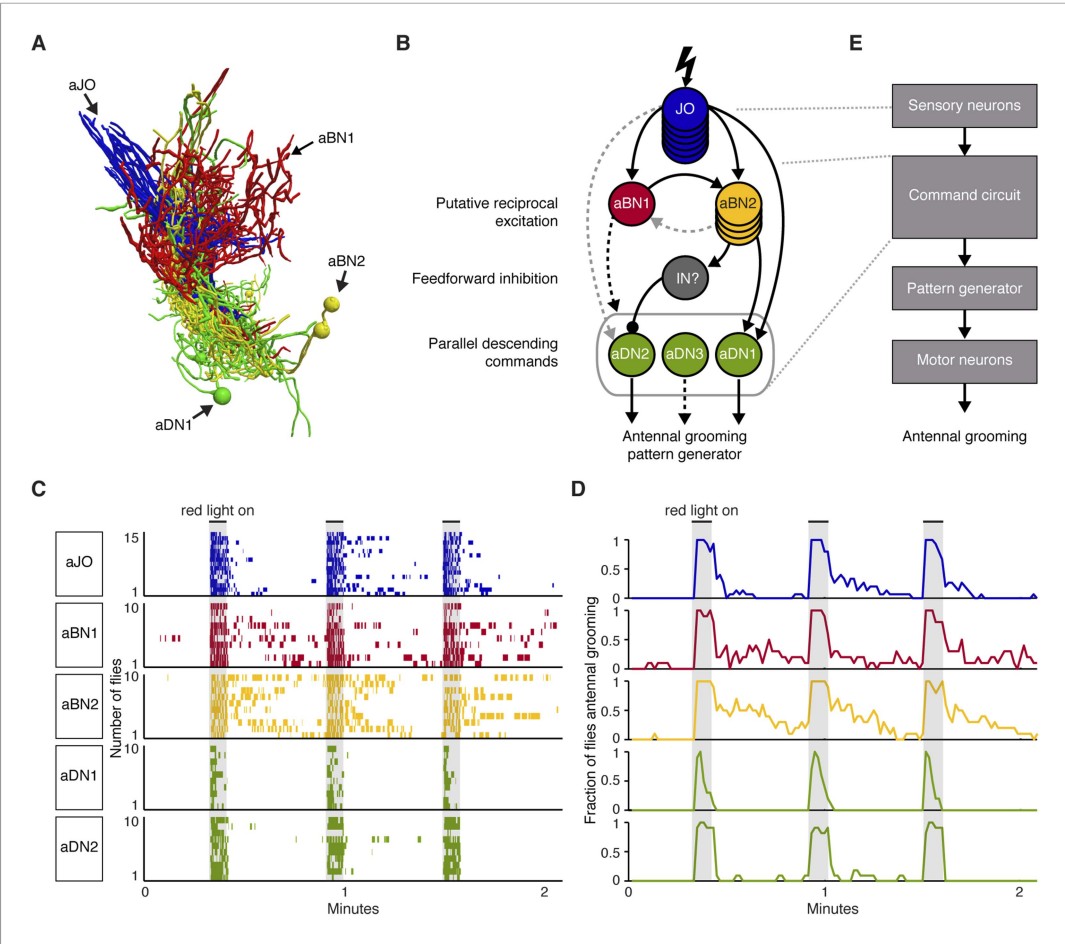

**Figure 7.** A circuit whose components elicit different antennal grooming durations. (**A**) The antennal grooming circuit (lateral view of tracings). Specific colors represent each neuron type shown in **B**. (**B**) Wiring diagram of the circuit. Lightning bolt represents mechanical stimulation of the antennae. Arrows represent excitatory cholinergic functional connections and the ball and stick indicates an inhibitory (picrotoxin sensitive) connection from an unidentified inhibitory neuron (IN). Note: JO neurons were previously reported to be cholinergic (*Yasuyama and Salvaterra, 1999*; *Salvaterra and Kitamoto, 2001*). Arrow to the gray oval surrounding the aDNs indicates that aBN1 provides excitatory input for aDN, but it is not known for which aDN(s). Gray dashed arrows indicate relatively weak and/or inconsistent connections (JO to aDN2 and aBN2 to aBN1). Text on the left highlights putative circuit connectivity features. Dashed arrow from aDN3 depicts presumed descending command. (**C**) The different neuronal classes induce distinct grooming responses. Ethograms of manually scored video showing antennal grooming induced with red light sensitive CsChrimson expressed in different spGAL4 pairs (aJO-spGAL4-1, aBN1-spGAL4-1, aBN2-spGAL4-1, aDN1-spGAL4-1, and aDN2-spGAL4-2). Ethograms of individual flies are stacked on top of each other. The gray bars indicate presentation of red light. Colors correspond to the wiring diagram (**B**) and indicate which neuronal class expressed csChrimson. Control flies did not perform antennal grooming (*Figure 7—figure supplement 1*). See *Video 4*, *Video 5*, and *Video 6* for representative examples. (**D**) Histograms representing the fraction of flies that were performing antennal grooming in **C** within one-second time bins. (**E**) The proposed organization of antennal grooming circuitry.

The following figure supplement is available for figure 7:

**Figure supplement 1.** CsChrimson activation of different neuronal classes.

absence of stimulation (*Major and Tank, 2004*; *Li et al., 2006*). To test this, we exploited CsChrimson's tight temporal control to activate these neurons briefly with red light, and then examined the dynamics of the grooming responses after red light cessation. Activation of either aBN1 or aBN2 elicited long antennal grooming durations that could persist for tens of seconds beyond red

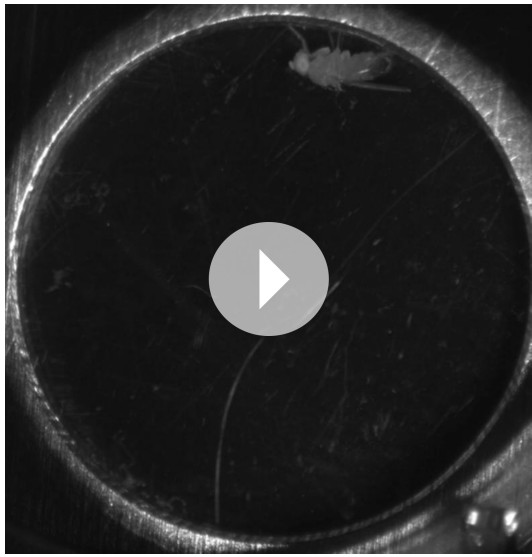

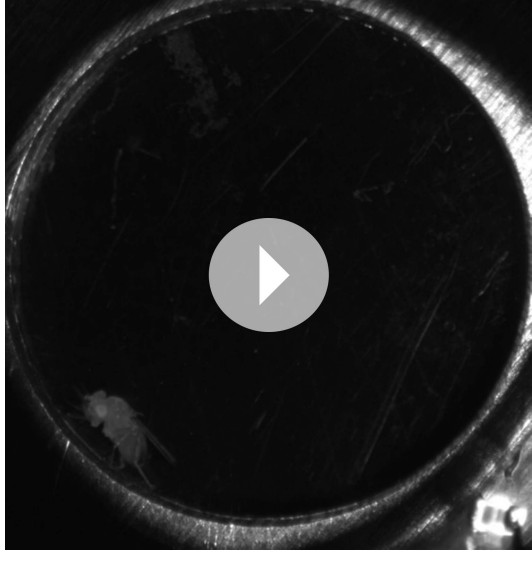

**Video 4.** Grooming in response to red light stimulation of CsChrimson-expressing aBN1 neurons. CsChrimson was expressed in aBN1 using aBN1-spGAL4-1. The infrared light in the bottom right hand corner shows when the red light was on to activate aBN1. Note that grooming persists upon cessation of the red light.

**Video 5.** Grooming in response to red light stimulation of CsChrimson-expressing aDN2 neurons. CsChrimson was expressed in aDN2 using aDN2-spGAL4-2. The infrared light in the bottom right hand corner shows when the red light was on to activate aDN2. Note that grooming does not persist upon cessation of the red light.

light cessation (*Figure 7C,D*, *Figure 7—figure supplement 1A*, *Video 4*). In contrast, activation of the aDNs, which are downstream of potential aBN-induced reciprocal excitation, should not induce persistent grooming (*Figure 7B*). Indeed, activation of aDN1 or aDN2 caused grooming that did not outlast the red light activation (*Figure 7C,D*, *Figure 7—figure supplement 1A*, *Video 5*). We observed similar results using additional spGAL4 pairs that target aBN1, aBN2, and aDN2 (not shown). Of note, activation of aDN2-elicited antennal grooming that lasted until the stimulus ended (*Figure 7C,D*), whereas activation of aDN1 elicited about 50% fewer grooming bouts that terminated prior to the red light turning off (*Figure 7C,D*, *Figure 7—figure supplement 1A,B*). This raises the intriguing possibility that the parallel aDNs induce different durations of antennal grooming. We next found that red light activation of the aJO, which is upstream of the aBNs, could also elicit grooming that persisted after red light cessation (*Figure 7C,D*, *Video 6*). This suggests that the aJO might excite the aBNs, which then induce persistent grooming. Taken together, our data indicate that the circuit produces persistent antennal grooming through aBN1 and aBN2. Future experiments will test whether the persistence is caused by reciprocal excitation or alternate mechanisms (see 'Discussion'). We concluded that our data describe an antennal grooming circuit whose components have the potential to modulate the duration of grooming, possibly mediated by their connectivity.

## Discussion

### Identification of a circuit that induces a specific grooming movement
This work provides the first description of a neural circuit that evokes a grooming movement: antennal displacements are detected by specific mechanosensory neurons that excite different interneuronal classes to elicit the front leg movements that constitute antennal grooming. We specifically identified JO neurons that project from the second antennal segment into the ventral brain, two brain interneuron classes (aBNs) that project within this ventral brain region, and descending neurons (aDNs) that project both within the ventral brain and descend to the VNS (*Figure 7A,B,E*). These neuronal classes are functionally connected with the common purpose of specifically inducing antennal grooming.

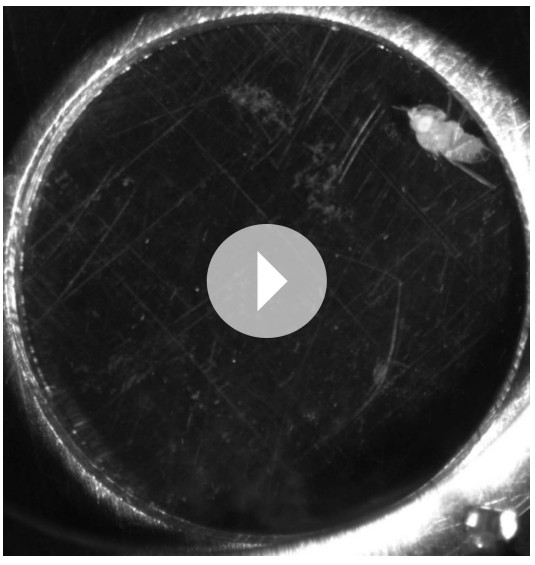

**Video 6.** Grooming in response to red light stimulation of CsChrimson-expressing aJO neurons. CsChrimson was expressed in the aJO using aJO-spGAL4-1. The infrared light in the bottom right hand corner shows when the red light was on to activate the aJO. Note that grooming persists upon cessation of the red light.

How does activation of this circuit elicit grooming of the antennae? Grooming movements are ultimately produced by pattern generators in the VNS that control the front and hind legs (*Berkowitz and Laurent, 1996*; *Burrows, 1996*). The aDNs project to the region of the VNS that should control front leg grooming movements, the ProNm. This suggests that the aDNs might connect to and activate neural circuits that generate coordinated antennal grooming movements (*Figure 7B,E*), but such circuitry remains to be identified. The tools for targeting aDNs acquired here may provide a means to identify this antennal grooming circuitry by examining their synaptic partners. Identification of this circuitry will provide a valuable example of how nervous systems achieve such exquisite specificity in targeting grooming movements to the site of a stimulus.

## Biological functions of the antennal grooming circuit

In this work, we examined a role for the identified circuit in detecting displacements of the antennae and eliciting grooming. This was based both on our discovery that the JO induces grooming, and that different JO neurons can detect antennal movements to elicit specific behavioral responses (*Kamikouchi et al., 2009*; *Yorozu et al., 2009*; *Matsuo et al., 2014*). Further, insects can encounter conditions in their natural environments, such as static electricity or unexpected mechanical disruptions that can move the antennae and cause different aversive behavioral responses (*Hunt et al., 2005*; *Newland et al., 2008*; *Jackson et al., 2011*; *Matsuo et al., 2014*). We have observed that flies show increased grooming of their antennae in chambers with high levels of static, and that this grooming was ablated using antistatic agents (unpublished observations). This led us to find that imposed deflections of the antennae induce grooming through the JO. Because such deflections likely disrupt normal antennal functions, such as hearing (*Göpfert and Robert, 2002*), grooming may provide a means of restoring them to their proper position and function.

Another function of grooming is to remove debris, parasites, or other substances from the body surface (*Sachs, 1988*; *Böröczky et al., 2013*; *Zhukovskaya et al., 2013*). We have shown that dust on the antennae induces grooming in flies (*Seeds et al., 2014*), however, it seems unlikely that small particles of dust could cause the large displacements that induce grooming in this work. This is supported by our observations that expression of TNT in the aJO does not cause defects in antennal dust removal (unpublished data), but does disrupt the grooming response to antennal displacements (*Figure 4E*). Therefore, dust might be detected by different sensory neurons, such as the mechanosensory bristle neurons found on the surface of the antennae. Bristle neurons are good candidates for stimulating grooming, given that mechanical stimulation of bristles on other body parts can induce site directed grooming responses (*Vandervorst and Ghysen, 1980*; *Corfas and Dudai, 1989*). Like JO neurons, the antennal bristle mechanosensory neurons send projections to the AMMC, and possibly to the SEZ (*Homberg et al., 1989*; *Melzig et al., 1996*), where they could induce grooming by interacting with the interneurons described in this work. This raises the possibility that multiple different sensory neuron types feed into the circuit to induce grooming. However, we have not yet identified spGAL4 lines that specifically target the antennal bristle mechanosensory neurons to allow us to test this hypothesis.

A scenario whereby different types of stimuli, such as antennal deflections or dust can cause grooming could explain some of the complexity of the circuit that we observe. Our data suggest

multiple possible routes of feedforward excitation through the circuit layers to induce antennal grooming. For example, the function of at least a subset of the aBN2 neurons was not necessary for grooming in response to physical displacement of the antennae, whereas they were necessary in response to thermogenetic activation of the JO. Thus, feedforward excitation likely has multiple routes through the antennal grooming circuit, but why would this be? One possibility is that each route reflects specific features of the sensory stimulus. Different stimuli might be sensed by distinct JO neurons or mechanosensory bristle neurons on the antennae that engage different ensembles of neurons within the circuit to produce specific grooming responses. Therefore, it may be revealing to test whether different stimuli coopt specific circuit components to fine–tune parameters of grooming, such as short vs persistent grooming durations.

The finding that the antennal grooming circuit can induce persistent grooming raises the question as to its biological role. Our results do not reveal a natural stimulus that induces persistence, as displacements of the antennae rarely induced persistent grooming (*Figure 4—figure supplement 3*). Persistence was previously described in the grooming responses of other animals, where it was found to be at least partially dependent on the nature of the stimulus (e.g., stimulus strength, duration, or frequency) (*Sherrington, 1906*; *Stein, 2005*). Thus, future work will explore the possibility that specific stimulus parameters can evoke persistent antennal grooming in flies. Additionally, persistence indicates the presence of a mechanism that produces long lasting neural activity within the grooming sensorimotor response. Such persistent activity could provide a mechanism by which temporal summation of successive stimuli is achieved. That is, successive, sub-threshold stimuli have been shown to be 'remembered' or summed to elicit grooming, although the neuronal mechanisms have not been identified (*Sherrington, 1906*; *Stein, 2005*; *Guzulaitis et al., 2013*). Therefore, future experiments will examine the role of specific features of the antennal grooming circuit in producing persistent grooming, and possible temporal summation (*Figure 7B*, possible circuit features discussed below).

## A subset of JO neurons detects antennal displacements and elicits grooming

aJO mechanosensory neurons are critical for the grooming response to antennal movements and sufficient to induce grooming. This reveals a new role for the JO, which was previously implicated in hearing, gravitaxis, and wind-induced suppression of locomotion (*Kamikouchi et al., 2009*; *Yorozu et al., 2009*). These behaviors are mediated by JO neurons that are localized to distinct regions of the mechanosensory structure and respond to high frequency vibrations (e.g., sounds) or more tonic movements (e.g., wind) (*Kamikouchi et al., 2006*; *Kamikouchi et al., 2009*; *Yorozu et al., 2009*; *Matsuo et al., 2014*). Thus, the JO is functionally organized such that different groups of mechanosensory neurons mediate different behaviors. Our finding that the JO elicits antennal grooming supports this idea, and demonstrates an even greater diversity of JO functions than previously thought.

Our data indicate that JO-induced antennal grooming is likely restricted to neurons projecting to zone C/E. Previously characterized JO neurons projecting to this zone are less sensitive to smaller movements of the antennae and adapt slowly to mechanical stimuli (*Kamikouchi et al., 2009*; *Yorozu et al., 2009*; *Matsuo et al., 2014*). The antennal displacement distance that induces grooming is within the range shown to induce calcium responses in previously described zone C/E neurons (*Yorozu et al., 2009*). This suggests that aJO neurons might also be activated by such large displacements, given that they are critical for the subsequent grooming response and project to zone C/E.

Two different populations of C/E-projecting neurons can induce antennal grooming; one that projects only to the AMMC, and the other (aJO) that projects to the AMMC and SEZ. This raises the question of whether both populations interact with the antennal grooming circuitry described in this work. The aBNs and aDN1 have neurites within the AMMC where they could plausibly receive excitatory inputs from both C/E neuron populations. Additionally, because neurons within zone C/E are implicated in other behaviors like wind-induced suppression of locomotion, it remains to be determined whether these JO neurons are multifunctional, or whether specific subpopulations within this group are responsible for distinct behaviors. Future work will be required to determine how these two populations of JO sensory neurons interact with the antennal grooming circuitry.

# A command circuit that elicits variable grooming durations

The antennal grooming circuitry consists of at least three different neuronal classes. As our extensive screening efforts identified multiple GAL4 lines for each class, we presume that we have uncovered a major portion of the neurons that elicit antennal grooming. Indeed, these neurons are sufficient to form a functionally connected circuit that extends from the JO to the VNS. However, additional descending neurons are likely involved given that TNT expression in the aDNs failed to disrupt the grooming response to aJO activation. We also found evidence for an unidentified neuron downstream of aBN2 that inhibits aDN2.

The neuronal classes in the circuit are each sufficient to elicit antennal grooming, similar to *command-like neurons* that evoke specific movements (also termed decision neurons or higher order neurons) (*Kupfermann and Weiss, 1978*; *Pearson, 1993*; *Kristan, 2008*; *Jing, 2009*). aBN1 and a subset of aBN2 neurons could potentially be more specifically termed *command neurons*, which are defined as being necessary and sufficient for initiating a specific movement, and fire in response to the movement-initiating sensory input (*Kupfermann and Weiss, 1978*). However, in order to definitively call these neurons command neurons, we need to test whether they are active in response to imposed movements of the antennae. Another term to describe collections of neurons that induce a behavior is a *command system* (*Kupfermann and Weiss, 1978*; *Jing, 2009*). However, a command system does not necessarily consist of functionally connected neurons. Therefore, given that these neurons constitute a functionally connected circuit, it may be appropriate to refer to them collectively as a *command circuit* for antennal grooming.

When the antennal grooming command circuit is compared with those that were previously identified in other systems, it emerges that circuits can consist of different layers that each elicit the specific movement, but with different durations (*Frost et al., 2001*; *Kristan et al., 2005*; *Pirri and Alkema, 2012*). In such cases, neurons have been found within particular layers that either elicit a movement that persists beyond their initial activation, or must be continually activated to continue the movement. These neuronal types have been termed trigger and gating neurons respectively (*Stein, 1978*). In the marine mollusc (*Tritonia diomedia*) and in the leech (*Hirudo medicinalis*), trigger neurons immediately downstream of sensory neurons induce persistent swimming motor patterns (named Tr1 in both animals) (*Frost et al., 2001*; *Kristan et al., 2005*). Gating neurons then induce swimming downstream of Tr1 in both animals (named DRI in the mollusc and 204/205 in the leech). This organization is strikingly similar to the layers of neurons that induce antennal grooming identified here; the aBNs induce persistent grooming and might be considered trigger neurons, whereas aDN2 could serve as a gating neuron. Taken together, the use of trigger and gating neurons might constitute a common organization in circuits that command specific movements downstream of a sensory stimulus.

The layered and complex organization of the antennal grooming circuit raises the question as to its function. We discussed above the possibility that the circuit may have different possible routes of feedforward excitation. Such organization may provide different points at which the circuit can be modulated, thus allowing for flexible control of different movement parameters. In the antennal grooming circuit, flexible control of movement duration could be provided by the parallel aDNs (*Figure 7B*), which appear to elicit different amounts of antennal grooming when activated. Artificially activated aDN1 elicits isolated grooming bouts that terminate despite continued activation, whereas activated aDN2 elicits grooming throughout the activation period. These observations are reminiscent of two descending neurons in molluscs that initiate biting movements during feeding, as one induces longer protraction durations whereas the other induces shorter durations (*Jing and Weiss, 2005*). This ability to generate distinct movement parameters, such as duration, may be a general feature of descending neurons that induce the same movement. Other parameters that might be differentially controlled are illustrated by locomotor systems, wherein descending neurons can elicit movement while controlling parameters such as speed and direction (*Dubuc et al., 2008*; *Roberts et al., 2010*; *Mullins et al., 2011*; *Puhl et al., 2012*; *El Manira and Grillner, 2014*).

The aDNs are also impinged upon by feedforward excitation and inhibition (*Figure 7B*), possibly to control which ones are active. For example, aBN2 provides differential control by exciting aDN1 and inhibiting aDN2. As there are multiple aDNs, it may be that feedforward excitation could impinge on them at the same time to induce hybrid durations of antennal grooming. Such an effect has been described for the two descending neurons controlling mollusc bite protraction, as activating the two

together produces intermediate durations (*Jing and Weiss, 2005*). Thus, we will further test whether the aDNs similarly produce flexible durations of antennal grooming.

The persistent grooming induced by brief excitation of the aBNs provides another mechanism for controlling grooming duration (*Figure 7B*). This is reminiscent of previously described grooming responses that were sustained despite stimuli cessation, which Sherrington referred to as afterdischarge (*Sherrington, 1906*; *Stein, 2005*). Persistence is not limited to grooming responses, as it has been described in behaviors as diverse as locomotion and courtship song (*Stein, 1978*; *Kristan et al., 2005*; *Inagaki et al., 2014*; *Gao et al., 2015*). In locomotion, reciprocal excitation among reticulospinal neurons has even been implicated in persistent tactile-induced swimming (*Li et al., 2006*). Although our work only provides weak evidence for reciprocal excitation between the aBNs, a mechanism whereby reciprocal connections maintain their excitation offers a plausible explanation for the persistence that we observe. Alternatively, aBN1 and aBN2 could have intrinsic membrane properties that allow them to produce prolonged responses to brief excitation (*Major and Tank, 2004*). Thus, future experiments will examine whether reciprocal excitation or intrinsic membrane properties of the aBNs produce persistent grooming.

In this section we have discussed how the complex organization of the antennal grooming circuit might control movement duration, however, it could also control parameters that our current analysis methods cannot detect. The level of behavioral analysis presented in this work reveals that the induced trajectories of the legs are specific for the antennae rather than other head parts; however, it does not allow for the detection of finer antennal grooming movements. Grooming is characterized by an initial targeting of the legs to the stimulated body part, followed by cyclic movements that groom the region (*Dürr and Matheson, 2003*). Higher resolution analysis of the leg kinematics could help resolve the boundary limits on the head to which the antennal grooming movements are confined during these two phases, and allow for determining how stereotyped these movements are. Furthermore, such analysis could elucidate how the legs interact with the antennal region to perform the grooming movements. Thus, higher resolution analysis would facilitate testing possible roles for the circuit in controlling different variables of antennal grooming, such as the limb trajectories or speed.

## Neurons that command specific movements in insects

Recent work indicates that command-like neurons may constitute a common means of eliciting specific movements in insects. Tools that allow for both acute control of neuronal activity and precise genetic targeting of specific neurons in fruit flies have enabled experiments that merge behavioral analyses with real time neuronal manipulations. Such experiments demonstrate that activation of specific neurons can elicit distinct movements, such as escape, locomotion, or courtship song (*Lima and Miesenböck, 2005*; *von Philipsborn et al., 2011*; *Gao et al., 2013*; *Flood et al., 2013b*; *Inagaki et al., 2014*; *Bidaye et al., 2014*; *von Reyn et al., 2014*). Additionally, work in crickets, grasshoppers, and locusts has revealed specific neurons that elicit courtship stridulation or flight (*Pearson et al., 1985*; *Hedwig, 1994*, *1996*, *2000*). Given our discovery of a circuit that specifically elicits antennal grooming, it would appear that the use of dedicated neurons to command specific movements is a common mechanism of behavioral control in insects. However, these command-like neurons may be embedded in larger, more dynamic neural networks in which neurons that elicit one movement also participate in other movements (*Kupfermann and Weiss, 1978*; *Kristan, 2008*). For example, in the leech several of the neurons involved in commanding swimming were found to also be excited during stimulus-induced shortening movements (*Shaw and Kristan, 1997*). Such findings indicate that neurons that can command one movement might also participate in the production of other movements, with their specific output subject to the collective activities of multiple different neurons within a given network (*Kristan et al., 2005*; *Kristan, 2008*). Thus, future experiments that examine whether the antennal grooming command circuit participates in controlling additional movements, may further elucidate the properties of neural networks that enable the performance of specific movements.

## Materials and methods

### Fly strains and rearing conditions

Flies were reared on cornmeal and molasses food at 21°C and 50% relative humidity on a 16/8 hr light/dark cycle. 5–8 day old males were used for all experiments, except for those in *Figure 6* that were

performed with 2–8 day-old flies. Stocks used in this study are listed in *Supplementary file 1* and *Supplementary file 3*.

## Identification of genetic reagents targeting neurons that induce antennal grooming

Three GAL4 lines that induced antennal grooming with thermogenetic activation using dTrpA1 were identified by screening over 1500 randomly selected GAL4 lines (R39A11, R26B12, and R18C11-GAL4) (*Seeds et al., 2014*). To identify additional lines, we visually screened through an image database of GAL4 expression patterns (*Jenett et al., 2012*) for those with expression in neurites close to aJO projections. Selected lines were crossed to *UAS-dTrpA1* and screened for increased antennal grooming at 30–32˚C. For pattern refinement and co-expression studies, the enhancers of GAL4 lines that exhibited increased antennal grooming were used to generate spGAL4 and LexA reagents, which were constructed as described previously (*Pfeiffer et al., 2010*) and produced by Gerald Rubin's lab. DBDs were inserted into the attP2 landing site (on chromosome 3), ADs were inserted into attP40 (on chromosome 2), and LexAs were inserted into attP40. *Supplementary file 1* lists the enhancer identities (*Jenett et al., 2012*) used to generate the spGAL4, LexA, and GAL4 lines. Control flies contain the DNA elements used for generating the different GAL4, spGAL4 halves, or LexA collections, but lack enhancers to drive their expression (images of control lines crossed to *UAS-GFP* are shown in *Figure 1—figure supplement 1E,F*). R27H08 (aJO-LexA) was identified in a screen of existing LexA lines (*Pfeiffer et al., 2010*) that targeted the aJO and could induce antennal grooming (*Figure 4—figure supplement 1A,B*).

## Behavioral analysis procedures

The camera setup and methods for recording the behavior of flies expressing dTrpA1 in different neuronal classes were described previously (*Seeds et al., 2014*). Behavior was recorded at 35 frames per second for 2 min at 30–31˚C. For amputation experiments, the entire antennae of 2 day-old males were severed with forceps and flies were allowed to recover for 4 days. The amputation did not damage the rest of the head or lead to mortality of the animals. CsChrimson experiments were performed in the dark, and flies were visualized for recording using an 850-nm infrared light source at 2 mW/cm$^2$ intensity (Mightex, Toronto, CA), which flies cannot see. For CsChrimson activation, we used 656-nm red light at 27 mW/cm$^2$ intensity (Mightex). The red light stimulus parameters were delivered using a NIDAQ board controlled through Labview (National Instruments, Austin, TX). Red light frequency was 5 Hz for 5 s (0.1-s on/off), and 30-s interstimulus intervals (total of 3 stimulations). Grooming movements were manually scored as previously described (*Seeds et al., 2014*), with the exceptions listed below. Manual scoring of prerecorded video was performed with VCode software and the data was analyzed in MATLAB (MathWorks Incorporated, Natick, MA).

Modifications of previously described scoring of grooming movements: *Antennal grooming:* The legs grasp and brush the antennae, with the head often tilted forward. When the antennae were amputated, 'antennal grooming' was scored when the legs were directed towards the area where antennae used to be. *Proboscis grooming:* The legs sweep down the proboscis when it is extended, or the tip when it is retracted. *Ventral head:* Legs sweep the area between the antennae and the ventral side of the head, as well as below the eyes towards the ventral bottom of the head. *Leg rubbing:* This movement was not included in this study; however, leg rubbing was often associated with the activated movements. Movement start times were scored one frame before the specific body part was first touched and ended two frames after that body part was last touched. The time interval between the previous and the next movement was scored as standing.

## Antennal displacement assay

Third antennal segments were coated with a mixture of iron powder (Atlantic Equipment Engineers, Upper Saddle River, NJ, 325 mesh) and UV cured glue (Kemxert Corp., York, PA) and then left to recover for 12 hr. The presence of the powder on the antennae did not cause increased antennal grooming after the recovery period. Flies were tethered using a pin that was glued to their thoraces and then positioned on a 6 mm diameter air-supported ball (*Seelig et al., 2010*) within a custom made electromagnet (*Figure 4C,D*, *Supplementary file 4*). The automated stimulation parameters of the electromagnet (frequency, on/off durations) were delivered via a NIDAQ board controlled through

MATLAB. Voltage was controlled using a Power Supply (B&K Precision Corporation, Yorba Linda, CA). The magnetic field was applied to move the antennae while the induced movements were recorded. Movement responses were manually scored as described above. The percent time the flies spent grooming their antennae was calculated through the duration of the experiment.

The stimulus parameters used for our experiments were determined using control flies ($w^{1118}$; *UAS-TNT; pBPGAL4U*). The percent time that control flies groomed when the magnetic field was applied was measured at different frequencies and voltages to determine the optimal stimulus conditions (*Figure 4—figure supplement 3A,B*). Flies performed an intermediate amount of grooming at 1 Hz and 10 V. Based on these results, antennal deflection experiments were performed at 1 Hz for 10 s (0.5 s on/off) at 10 V, with a 30 s wait time between stimulations (total of 4 stimulations). The magnetic field strength at 10 V was measured at 570 Gauss. Grooming was not induced with flies that lacked iron powder on their antennae and were exposed to the magnetic field (data not shown). The antennal displacement caused by the magnetic field was measured by recording video of the antenna from a side view (*Figure 4—figure supplement 3C*). Antennal displacement was calculated by measuring movement of the distal tip of the third antennal segment before and after the magnetic field was applied. Pixels were calibrated to physical distances using a known standard.

## Statistical methods
Behavioral data was analyzed with nonparametric statistical tests. First, we performed a Kruskal–Wallis (ANOVA) test to compare more than three genotypes with each other. Next we performed a post-hoc Mann–Whitney U test and applied Bonferroni correction.

## Immunostaining and image analysis
Dissection and staining was performed as previously described (*Hampel et al., 2011*), with modifications listed below. 5–8 day-old males were used for all dissections, except for the stochastic labeling experiments using MCFO-1 (*Nern et al., 2015*) in *Figure 1—figure supplement 4A–H*, where 1–2 day-old flies were dissected. For *Figure 1E,F*, *Figure 2C–F*, *Figure 3A–D*, *Figure 5A–H*, *Figure 1—figure supplement 1F*, *Figure 1—figure supplement 2A–E*, *Figure 2—figure supplement 2A–I*, *Figure 4—figure supplement 1B–D*, *Figure 4—figure supplement 2A–F*, and *Figure 5—figure supplement 2A–H* additional treatment was performed to clear the tissue: After immunohistochemistry, tissue samples were post-fixed in 4% paraformaldehyde in PBS for 4 hr at room temperature, followed by four 30 min washes in PBT. Before mounting the CNS on a poly-L-lysine (P1524; Sigma, St. Louis, MO) coated cover slip, the tissues were washed in PBS for 15 min to remove Triton. After mounting the tissues on the poly-L-lysine-coated cover slip, they were dehydrated through a series of ethanol dilutions (30%, 50%, 75%, 95%, and 3 × 100%) for 10 min each, followed by an incubation series in 100% xylene (Fisher Scientific, Fair Lawn, NJ) three times for 5 min each in Coplin jars. Afterwards tissues were embedded in DPX, a xylene-based mounting solution (Electron Microscopy Sciences, Hatfield, PA, Cat#13512) and allowed to dry for 48 hr before imaging.

For *Figure 2H*, *Figure 1—figure supplement 1C–E,G–J*, *Figure 1—figure supplement 3A–G*, *Figure 1—figure supplement 4A–H*, *Figure 2—figure supplement 1B–H*, and *Figure 5—figure supplement 1A–I* stained tissue samples were mounted after the immunohistochemistry in Vectashield (Vector Laboratories, Inc. Burlingame, California) and allowed to incubate for an hour before imaging.

Antibodies used: rabbit anti-GFP (1:500, Thermo Fisher Scientific, Waltham, MA, #A11122), chicken anti-GFP (1:2000, Abcam, Cambridge, MA, #ab13970), mouse anti-GFP (1:200, Sigma, #G6539), mouse mAb anti-nc82 (1:50, Developmental Studies Hybridoma Bank, University of Iowa), rat anti-DN-cadherin (1:20, Developmental Studies Hybridoma Bank), rabbit anti-RFP (to detect tdTomato; 1:1000, Clontech Laboratories, Inc., Mountain View, CA, #632496), rat anti-flag (Novus Biologicals, LLC, Littleton, CO, #NBP1-06712), rabbit anti-HA (Cell Signaling Technology, Danvers, MA, #3724S), mouse anti-V5 (AbD Serotec, Kidlington, England, #MCA1360), AlexaFluor-488 (1:500; goat anti-rabbit, goat anti-chicken, goat anti-mouse; Thermo Fisher Scientific), AlexaFluor-568 (1:500; goat anti-mouse, goat anti-rat; Thermo Fisher Scientific), AlexaFluor-633 (1:500; goat anti-rat; Thermo Fisher Scientific).

Confocal stacks of stained CNS and antennae were imaged on a Zeiss LSM710 confocal microscope with a Plan-Apochromat 20×/0.8 M27 objective and a Plan-Apochromat 63×/1.4 oil immersion objective. To visualize the neuronal classes together as shown in *Video 2*, confocal images of different split-GAL4 lines were computationally aligned from individual specimens to one brain sample of our own collection with the Computations Morphometry Toolkit CMTK (https://www.nitrc.org/projects/cmtk/) (*Jefferis et al., 2007*) and assembled in FluoRender, (*Wan et al., 2009*) a suite of tools for viewing and analyzing image data.

Image preparation, analysis of overlap, and adjustment of brightness and contrast were performed with Fiji (http://fiji.sc/). *Figure 5* shows maximum projections of confocal stacks that were modified using the 3D viewer plugin in Fiji to crop contaminating neurons and background noise. The same images are shown in *Figure 5—figure supplement 2A–H* with only brightness and contrast adjusted. Confocal stacks of brains imaged with a 63× objective were used to reconstruct each neuronal class shown in *Figure 3E*. We traced each neuron from different brains with neuTube software (*Feng et al., 2015*) and assembled the neuronal circuit manually.

## GCaMP6s imaging and analysis of calcium responses

Male flies were dissected in saline containing 103 mM NaCl, 3 mM KCl, 5 mM TES, 8 mM trehalose dihydrate, 10 mM glucose, 26 mM NaHCO$_3$, 1 mM NaH2PO$_4$, 2 mM CaCl$_2$, 4 mM MgCl$_2$, and bubbled with carbogen. The brain and VNS were placed on a poly-lysine-coated coverslip. Dissections were visualized using minimal illumination to avoid activation of CsChrimson. The preparation was continuously perfused in saline at 60 ml/hr. Imaging was done using a two-photon scanning microscope (Bruker, Billerica, MA) with an excitation wavelength of 920 nm. Imaging fields of view were chosen as to contain a distinctive process of the candidate post-synaptic neuron, which included processes at the midline for the aDNs and aBN2, and vertical running processes close to the midline for aBN1.

CsChrimson was excited with 2 ms pulses of 590-nm light via an LED shining through the objective. Instantaneous powers measured out of the objective ranged between 50 µW/mm$^2$ and 800 µW/mm$^2$. Trains were delivered at 50 Hz. Experiments usually started with a 50-pulse train. If no response was observed, the power was raised progressively until one occurred or the maximum power was reached. Each experimental run consisted of 4 repeats lasting for approximately 20 s. Runs were repeated approximately every 2 min. When postsynaptic responses were observed, we did not see any desensitization. See *Supplementary file 2* for stimulus conditions used for data shown in *Figure 6*. For blocking nicotinic or GABAergic/glutamatergic transmission, mecamylamine (50 µM) or picrotoxin (30 µM) were administered through a perfusion line for 3–10 min, followed by a drug-free wash (drugs from Sigma). Pharmacology experiments were run using samples that showed consistent responses. Stimulus conditions were chosen that elicited reliable transient responses. Average traces shown in *Figure 6E–I* are from those trials where pharmacology experiments were done (trials used to generate average traces shown in *Figure 6—figure supplements 2, 3*). Average traces shown in *Figure 6A–D* correspond to all flies and trials at the indicated stimulus conditions shown in *Figure 6—figure supplements 2, 3*.

ΔF/F0 was calculated (F0 is the average signal before the stimulation) for each video in a region of interest (ROI) obtained as follows: the average projection of the video was calculated and pixels of the projection were clustered by a k-means algorithm between ROI and non-ROI pixels. Of note, the selection method relies only on average intensity and not activity, because we wanted to use the same detection method for responsive and non-responsive runs. This also relies on selecting fields of view that unambiguously contain only the neuron of interest. Analysis was run in Julia (http://julialang.org/).

## Acknowledgements

We thank: Heather Dionne, Barret Pfeiffer, and Gerald Rubin for providing spGAL4 lines and pJFRC26, pJFRC34, and pJFRC48 flies; Eric Hoopfer for MATLAB code used to analyze behavioral data; Karen Hibbard and Don Hall for stock-building support; Vivek Jayaraman for advice and reagents for calcium imaging experiments (author Romain Franconville is a member of the Jayaraman lab); Rebecca Johnston, Gudrun Ihrke, Jennifer Jeter, Gina DePasquale, and Joanna Hausenfluck for protocols and immunohistochemistry on selected spGAL4 lines; Jeff Jordan, Lakshmi Ramasamy, and Steven Sawtelle for constructing the electromagnet and optogenetic setups; Carmen Robinett, Claire

McKellar, Scott Sternson, and Bill Kristan for editing and comments on the manuscript; Emily Nielson for administrative support; TJ Florence for help with antennal displacement measurements. Jeremy Cohen for advice related to antistatic agents; The Howard Hughes Medical Institute supported this work.

## Additional information

### Funding

| Funder | Author |
|---|---|
| Howard Hughes Medical Institute (HHMI) | Julie H Simpson |

The funder had no role in study design, data collection and interpretation, or the decision to submit the work for publication.

### Author contributions

SH, AMS, Conception and design, Acquisition of data, Analysis and interpretation of data, Drafting or revising the article; RF, Performed calcium imaging experiments and analyzed the calcium imaging data, Drafting or revising the article; JHS, Conception and design, Drafting or revising the article

### Author ORCIDs

Romain Franconville, http://orcid.org/0000-0002-4440-7297
Andrew M Seeds, http://orcid.org/0000-0002-4932-6496

## Additional files

### Supplementary files

• Supplementary file 1. Enhancer identities used to target different neuronal classes.

• Supplementary file 2. Stimulus conditions used for the experiment shown in *Figure 6*.

• Supplementary file 3. Stocks used in this study.

• Supplementary file 4. Design plans for the fly electromagnetic stimulation rig.

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
