## [Decision Letter]

Thank you for submitting your work entitled “A neural command circuit for grooming movement control” for peer review at *eLife*. Your submission has been favorably evaluated by K VijayRaghavan (Senior Editor), a Reviewing Editor, and three reviewers.

The reviewers have discussed the reviews with one another, and the Reviewing Editor has drafted this decision to help you prepare a revised submission.

All reviewers agree that this is an extremely thorough characterization of the neural circuits underlying an interesting behavior in *Drosophila*. For example, one of the reviewers said: “The authors have done an impressive job in leveraging the state-of-the-art genetic/neural circuit tools in *Drosophila* to determine which neurons are necessary/sufficient for this behavior and how they functionally connect with each other. Moreover, they have a nicely supported model for functional interactions in the circuit, including the role of inhibition – this is terrific!”. Another reviewer commented: “This study will be highly interesting for neuroscientists in the field of motor control as well as for scientists studying animal behavior in general”. A third reviewer also stated: “The authors identify novel chordotonal sensory neurons innervating Johnston's Organ (JO) in the second antennal segment, and 3 classes of interneurons involved in this behavior, including 2 classes of local neurons (LNs) and several classes of descending neurons (DNs). This clearly required a heroic effort, and the authors are to be congratulated for their work”.

Remarkably, optogenetic activation of any of these classes of neurons drives grooming of the antenna, and not of other body parts (at least during photostimulation) (Figure 7—figure supplement 1). This observation offers a fascinating insight into the fundamental question of how the fly's brain directs its appendages to groom the precise part of the body surface from which it is receiving a sensory irritant or input: evidently the sensory neurons that innervate that part of the body surface activate a motor circuit that controls limb movement with precisely the right trajectory to direct them to the location from which the sensory input was received. The data presented here show that deeper layers of the circuit exhibit a similar level of functional specificity. It is almost as if this part of the fly's sensory-motor system is built like a marionette: a dedicated, position-specific sensory-interneuron-motor loop for each body part that needs to be groomed. While there were hints of this in the original paper, the present study demonstrates this much more clearly and the degree of specificity uncovered, at all layers of the circuit, is remarkable. Although the results of this paper do not yet explain how this sensory information is converted to such a specific limb trajectory through 3-D space at the motor level, they provide an important step in this direction. It is, therefore, surprising that the authors do not raise this interesting point at all in their Introduction and Discussion. They are encouraged to do so.

In summary, the authors have taken an important step forward in deconstructing a circuit that may ultimately explain a fascinating problem in sensory-motor coordination. They are encouraged to downplay the reciprocal excitation/persistent activity part of their story, which is the weakest, and emphasize the aspects of their findings that are most relevant to the grooming system, namely the remarkable functional specificity they have uncovered. The circuit architecture they have defined certainly hints at a more complex function and regulation than would be expected from a simple feed-forward stimulus-response reflex pathway, and this is interesting as well. Above all, they should make clear to the reader (especially the non-specialist) the limitations of the methods they have employed, and the parts of the story that are rock-solid vs. those that are less so.

Essential revisions:

1) Which sensory neurons of the JO induce grooming?

1A) It is unclear what distinguishes antennal movements for song, wind, and gravity perception (what these antennal movements are known to be important for) and antennal movements for grooming. Is it a much larger displacement of the antenna? Or is it the context? The authors show a plot where they increase the strength of the magnetic field (measured in volts) and show more grooming. They could easily measure the displacement of the antenna (using high speed video or a laser doppler vibrometer) and determine what range of antennal displacements (measured in nm) leads to grooming – this information is critical for knowing what strength of the stimulus activates the antennal grooming circuits. This is important because the receptor neurons for their behavior are likely overlapping with the receptor neurons for song, wind, and gravity detection, as the authors mention.

1B) The authors interpret their findings to that effect that a specific subpopulation of JO neurons is exclusively responsible for providing sensory information specifically for antennal grooming. I agree that activation of these sensory neurons (SN) shows that they are sufficient to induce grooming. This, however, might be a general and unspecific stimulation of the JO which, in turn, elicits a general grooming response. You show in Figure 4 that silencing aJO neurons significantly reduces grooming; while this suggests that aJO is involved in eliciting grooming it only shows sufficiency, not necessity. Also, this is still consistent with a general decrease of JO sensitivity. Therefore, the following questions need to be addressed to support the conclusion that these SNs are specific for grooming: (i) did activation of the other previously described subpopulations of JO SNs (e.g. those described to be specific for hearing, gravitaxis, or wind sensitivity) fail to induce grooming? (ii) if aJO neurons are indeed the sole JO neurons necessary for grooming then ablation of all non-aJO neurons should not result in a decrease in grooming behavior; (iii) it may have slipped my attention, but was the role of antennal exteroceptors, e.g. tactile hairs, tested? All of these points have to be addressed before you can definitely say that aJO neurons are necessary and sufficient for grooming, i.e. part of the grooming circuit. As it stands now, you have shown that grooming can be induced by stimulating at least part of the JO.

1C) The authors show that the activity of several (but not all) of the classes of neurons they have described is required for this behavior. That is reassuring, at one level. However inspection of the video suggests that the magnitude of the antennal displacements (while not explicitly measured) produced by the magnet must be far greater than those normally produced by the application of dust. This raises the important question of whether the neurons described here are required for dust-evoked grooming. The authors do not comment on this point, and should. Presumably, they have tried this experiment, and apparently it did not work (otherwise why would they have gone to the trouble of constructing the elaborate magnet-evoked assay). The authors need to be up front with their readers about this, raise the issue and discuss it. There may be an interesting reason for the discrepancy. For example, is the artificial stimulus used here actually mimicking the antennal displacements produced by grooming behavior itself, rather than by the small particles that typically evoke grooming? If so, could the circuit uncovered here be part of a mechanism for feedback-control of grooming behavior? There is some hint of this, briefly touched on by the authors in passing, because amputation of the antenna actually causes an increased level of grooming behavior produced by thermogenetic activation of some of the interneuron classes. In any case, in the interests of full disclosure the authors need to discuss this important issue.

2) The Introduction is not broad enough. For example, the authors need to provide more background on CPGs, patterned motor behaviors, and grooming, and to place the behavior they are studying in the context of the larger body of work. For example, the authors only cite papers on stick insect walking when introducing stereotyped motor patterns – they need to invoke behaviors like song production, feeding, etc. and compare/contrast their behavior with other well-studied motor patterns. In addition, the authors also don't provide enough background on their previous study (69) – this study is in many ways a follow-up, but they expect the reader to have familiarity with this paper already.

3) Where do individual aJOs project? The authors don't report on any single cell clones to show that individual aJOs innervate all 3 brain regions mentioned (AMMC, pSEZ and vSEZ) or that individual aJOs only innervate one region. If this experiment was not possible for some technical reason can the authors indicate this in the text of the manuscript?

4) Functional connectivity versus synaptic connectivity: The data presented clearly show functional connectivity between neurons. However, there is only weak evidence (it is not ‘likely’), that functionally connected neurons are postsynaptic to each other since synaptic delays are known to be small relative to the temporal resolution of the GCaMP6s used. Their approach detects functional connectivity but not synaptic connectivity – which is fine. However, the authors should make this clear throughout the manuscript (and in the Abstract).

5) Resolution of the behavior: Given the authors' approach to scoring the behavior manually (and lumping all of antennal grooming into one category) it is not possible to examine whether activity of different neurons in the antennal grooming circuit leads to different types/aspects of behavior or different timing of the behavior or differences in trial-to-trial reproducibility. Given that the authors suggest in their Introduction and Discussion that this behavior is a good model for understanding the contribution of sensory drive and modulation to stereotyped motor programs, it would help to know how stereotyped the behavior is. The authors should add a paragraph to their Discussion of the future value of examining the grooming behavior of *Drosophila* at higher resolution (think Dragonfly or *Drosophila* flight/steering, *Drosophila* or bird song production, ant path navigation, or arm reaching). In other words, there is much more to get out of this system now that the circuit is mapped.

6) The differentiation of the classes of antennal grooming descending neurons (aDN) appears to be less specific compared to the two aLN-classes. For example, no information is given on aDN2 in Figure 3 and its legend. In general throughout the text the number of DN-classes that are referred to is changing from two (in the subsection “Three different interneuron classes elicit antennal grooming”), to one (in the subsection “A putative circuit that elicits grooming in response to antennal displacement”) and to three (also in the aforementioned subsection). In order to justify the summary Figure 7, it is important that the authors allow the reader to keep clear track for all data available on DN classes leading to this scheme.

7) In the third paragraph of the subsection “Connectivity among the antennal grooming neuronal classes”, based on the neuroanatomical analysis of the branching patterns of the described sensory and brain interneurons, the authors conclude specific connectivity between them. I agree that their data show that these neurons are likely to belong to the same circuit, but I do not see proof of an explicit synaptic connectivity between them: (i) the anatomical analysis of the branching patterns has not the scale to allow for that – a scale of 25µm as given in the figures is not sufficient to resolve connectivity (e.g. see Figure 5) and (ii) the same holds for the calcium transients recorded (e.g. see Figure 6). Such explicit interpretation is not justified. A terminology like “neurons downstream” or “neurons upstream” is appropriate.

8) With regards to the subsection “A command circuit that elicits variable grooming durations”, all neural circuits are composed of “multiple neurons”. It is not clear which point the authors want to make. Specifically, information is lacking on how the new findings relate to previous findings on neurons that serve command-like functions in insect behavior (work on courtship behavior in locusts and crickets; mostly Hedwig and coauthors) and in the fruit fly? With respect to the fruit fly I was missing a discussion on what the new study adds to the previous work on command neurons in the control of fruit fly courtship behavior (e.g. [78]), aversive motor behavior (19) and walking direction (2)?

9) Is there really a reciprocal excitation loop in this circuit, and is it responsible for persistent grooming behavior evoked by optogenetic stimulation? The authors place a great deal of emphasis on the finding of an apparent reciprocal excitation loop between the aLN1 and aLN2 local interneurons (Figure 7), and invoke it to explain their interesting observation that activation of these interneurons can produce persistent grooming (Figure 7). Unfortunately, due to limitations in the specificity of the reagents used to perform these experiments, the evidence of reciprocal excitation is unconvincing and it is very difficult to correlate the physiological (imaging) and behavioral data. Firstly, the evidence of reciprocal connectivity is weak. The degree of anatomical overlap between aLN1 and aLN2 neurons is very limited (Figure 5 – by the way, please indicate with dashed outlines the location of the AMMC, posterior and ventral SEZ on these figures for the readers' benefit). Although the evidence for functional connectivity from LN1→LN2 neurons is convincing (Figure 6), the evidence for the reciprocal connection (Figure 6) is not. Apparently (based on the information provided in Figure 6—figure supplement 3), to test for the latter connection the authors used an LN2-LexA driver to express CsChrimson in these neurons (in contrast to the reverse experiment where they used an LN1-GAL4 driver to express the opsin). This LN2-LexA driver is extremely broadly expressed, in the central brain and optic lobes (Figure 4—figure supplement 1). Without specific 2-photon targeted activation of the LN2 neurons themselves, it is difficult to be confident that the excitatory drive on the LN1 neurons in Figure 6 is indeed due to input from the aLN2 cells.

10) Furthermore, the magnitude of the average LN1 response to “LN2 stimulation” in Figure 6 is much weaker than that of the LN1 → LN2 stimulation in Figure 6. Close inspection of the supplemental data indicates that even this weak average recording is misleading, because only a fraction of the flies tested, and only a small fraction of trials, yielded any response at all (Figure 6—figure supplement 3, LN2 to LN1). This is very different from a result in which the neurons showed a weak response in most of the flies in most of the trials. This is all the more puzzling given that there are more aLN2 than aLN1 cells (Figure 7), and that the activation of neurons in the aLN2-spGAL4-2 line has the strongest effect to promote grooming of any of the lines characterized in the paper (Figures 2 and 7). Thus, the evidence for an LN2→LN1 connection that is relevant to grooming is, in my opinion, weak and unconvincing. At the very least, the circuit diagram in Figure 7 should be re-drawn to show that the connection from aLN2→aLN1 is far weaker and more questionable than the reverse connection (perhaps using a dashed line and question mark), and this issue and associated caveats should be clearly spelled out in the Discussion.

11) In addition, if the authors really want to make the case that this putative reciprocal excitatory loop is responsible for the persistent grooming behavior they observe, then they should perform an epistasis experiment in which they demonstrate that persistent grooming triggered by aLN2 activation (but perhaps not grooming per se) is blocked by inactivating aLN1 cells, and/or vice-versa.

Finally, the authors should point out that in their antennal displacement assay, only a small fraction of stimulation trials elicited persistent grooming, and that even in those cases the persistence appears to be in the grooming of the eyes/head, not the antenna (Figure 4—figure supplement 3). So the biological relevance of this persistence phenomenon, interesting though it may be, is not obvious.

---

## [Author Response]

*[…] In summary, the authors have taken an important step forward in deconstructing a circuit that may ultimately explain a fascinating problem in sensory-motor coordination. They are encouraged to downplay the reciprocal excitation/persistent activity part of their story, which is the weakest, and emphasize the aspects of their findings that are most relevant to the grooming system, namely the remarkable functional specificity they have uncovered. The circuit architecture they have defined certainly hints at a more complex function and regulation than would be expected from a simple feed-forward stimulus-response reflex pathway, and this is interesting as well. Above all, they should make clear to the reader (especially the non-specialist) the limitations of the methods they have employed, and the parts of the story that are rock-solid vs. those that are less so*.

We thank the reviewers for their kind and thoughtful comments regarding this work. The manuscript is revised to better emphasize the discovery of a circuit that induces a highly specific grooming movement. The possible role of reciprocal excitation as the cause of persistent grooming is now deemphasized, but the manuscript retains some focus on the circuit as a potential mediator of grooming duration. We have added additional experiments related to the antennae and Johnston’s organ requested by the reviewers, including two full figure supplements for Figure 1 (Figure 1—figure supplement 3 and Figure 1—figure supplement 4), and an additional panel for Figure 4—figure supplement 3. We revised the Abstract, and the Introduction and Discussion are expanded to include topics that the reviewers requested. Finally, we have renamed the interneurons aLN1 and aLN2 to aBN1 and aBN2 as suggested by the reviewers. Our responses to each requested revision are listed below.

*Essential revisions*:

1) Which sensory neurons of the JO induce grooming?

Responses to specific queries for question 1 are addressed independently (i.e. 1A, 1B (parts i, ii, iii), and 1C). At the end, these responses are assembled into a cohesive conclusion. Additionally, because question 3 also fits nicely with this question, we incorporated our response to it with question 1 conclusions.

1A) It is unclear what distinguishes antennal movements for song, wind, and gravity perception (what these antennal movements are known to be important for) and antennal movements for grooming. Is it a much larger displacement of the antenna? Or is it the context? The authors show a plot where they increase the strength of the magnetic field (measured in volts) and show more grooming. They could easily measure the displacement of the antenna (using high speed video or a laser doppler vibrometer) and determine what range of antennal displacements (measured in nm) leads to grooming – this information is critical for knowing what strength of the stimulus activates the antennal grooming circuits. This is important because the receptor neurons for their behavior are likely overlapping with the receptor neurons for song, wind, and gravity detection, as the authors mention.

We measured how far the antennae are displaced when the electromagnet was driven at two different voltages (5 and 10V). Five volts induces limited grooming whereas ten volts induces near the maximum amount of grooming (Figure 4—figure supplement 3). The measured displacements at these voltages are 11 +/- 3 μm (5V) and 31 +/- 6 μm (10V) (new data added as Figure 4—figure supplement 3). The displacement distance at ten volts is within the range shown to induce calcium responses in JO neurons projecting to zones C/E (81). Interestingly, aJO neurons identified in this work project to zone C/E (new data added as Figure 1—figure supplement 3), suggesting that they too might be activated by such displacements. However, because these results demonstrate that the amplitude of the stimulus is capable of inducing previously described zone C/E neurons, this experiment does not resolve how the stimulus induces grooming behavior versus wind-induced suppression of locomotion.

The measurement of antennal displacement magnitude is now in the Results text and incorporated into Figure 4—figure supplement 3. Additionally, we mention in the Discussion that this displacement magnitude is likely to cause calcium responses in previously described zone C/E neurons and provide limited speculation on the implications. Finally, text was added to the Materials and methods that describe how we measured the antennal displacements.

*1B) The authors interpret their findings to that effect that a specific subpopulation of JO neurons is exclusively responsible for providing sensory information specifically for antennal grooming. I agree that activation of these sensory neurons (SN) shows that they are sufficient to induce grooming. This, however might be a general and unspecific stimulation of the JO which, in turn, elicits a general grooming response. You show in*
Figure 4
*that silencing aJO neurons significantly reduces grooming; while this suggests that aJO is involved in eliciting grooming it only shows sufficiency, not necessity. Also, this is still consistent with a general decrease of JO sensitivity. Therefore, the following questions need to be addressed to support the conclusion that these SNs are specific for grooming: (i) did activation of the other previously described subpopulations of JO SNs (e.g. those described to be specific for hearing, gravitaxis, or wind sensitivity) fail to induce grooming?*

We tested whether activation of previously described JO subpopulations induces grooming. Excitation of zone C/E neurons using previously published GAL4 drivers elicits antennal grooming, whereas excitation of zone A/B neurons did not (new data added as Figure 1—figure supplement 3). We added a co-stain of aJO with two previously described C/E GAL4 lines to show that aJO neurons described in this study also project to zone C/E (new data added as Figure 1—figure supplement 3, also discussed above in reference to reviewers' comment 1A). Taken together, our data demonstrates that a subset of JO neurons (zone C/E) are involved in inducing antennal grooming. The fact that activation of neurons projecting to other zones does not induce grooming argues against the possibility that antennal grooming is a non-specific effect of generally activating any JO neurons.

*(ii) If aJO neurons are indeed the sole JO neurons necessary for grooming then ablation of all non-aJO neurons should not result in a decrease in grooming behavior*.

Our results indicate that expression of TNT in the aJO almost completely blocks the grooming response to imposed movements of the antennae (Figure 4). Therefore, we described these neurons as being necessary for this response. We believe that calling aJO neurons necessary does not imply that they are the only sensory neurons on the antennae that induce grooming in response to antennal deflections. However, we changed two sentences in the manuscript to now state that the aJO neurons are critical rather than necessary. A sentence in the third paragraph of the subsection “A putative circuit that elicits grooming in response to antennal displacement” now states: “The aJO was critical for this response because expression of TNT in these neurons significantly reduced antennal and head grooming in response to the magnetic field (Figure 4, blue boxes, head grooming not shown).” The first sentence in the subsection “A distinct class of JO neurons detects antennal displacements and elicits grooming” now reads: “aJO mechanosensory neurons are critical for the grooming response to antennal movements and sufficient to induce grooming.”

*(iii) It may have slipped my attention, but was the role of antennal exteroceptors, e.g. tactile hairs, tested? All of these points have to be addressed before you can definitely say that aJO neurons are necessary and sufficient for grooming, i.e. part of the grooming circuit. As it stands now, you have shown that grooming can be induced by stimulating at least part of the JO*.

The bristle mechanosensory neurons on the antennae have not been tested for their role in grooming because we have not yet acquired spGAL4 lines that can target them. We added text to address the bristle mechanosensory neurons in the Discussion (for further information, see our comments below for 1C).

1C) The authors show that the activity of several (but not all) of the classes of neurons they have described is required for this behavior. That is reassuring, at one level. However inspection of the video suggests that the magnitude of the antennal displacements (while not explicitly measured) produced by the magnet must be far greater than those normally produced by the application of dust. This raises the important question of whether the neurons described here are required for dust-evoked grooming. The authors do not comment on this point, and should. Presumably, they have tried this experiment, and apparently it did not work (otherwise why would they have gone to the trouble of constructing the elaborate magnet-evoked assay). The authors need to be up front with their readers about this, raise the issue and discuss it. There may be an interesting reason for the discrepancy. For example, is the artificial stimulus used here actually mimicking the antennal displacements produced by grooming behavior itself, rather than by the small particles that typically evoke grooming? If so, could the circuit uncovered here be part of a mechanism for feedback-control of grooming behavior? There is some hint of this, briefly touched on by the authors in passing, because amputation of the antenna actually causes an increased level of grooming behavior produced by thermogenetic activation of some of the interneuron classes. In any case, in the interests of full disclosure the authors need to discuss this important issue.

We discuss this in the first two paragraphs of a new subsection of the Discussion called “Biological functions of the antennal grooming circuit*.*” In brief, the large deflections that induce grooming described in this paper are likely to be caused by non-debris stimuli such as static electricity or unexpected displacements of the antennae. We cite several papers that support this idea. We also address the role of this circuitry in the removal of debris, such as dust from the antennae. Different sensory neuron types other than the JO neurons may play a role in detecting debris on the antennae and inducing grooming (e.g. antennal mechanosensory bristles). However, we have not yet found lines that specifically target the bristle neurons to test this possibility.

*3) Where do individual aJOs project? The authors don't report on any single cell clones to show that individual aJOs innervate all 3 brain regions mentioned (AMMC, pSEZ and vSEZ) or that individual aJOs only innervate one region*. *If this experiment was not possible for some technical reason can the authors indicate this in the text of the manuscript?*

We added a multicolor stochastic labeling experiment to the manuscript. This revealed that the majority of neurons within the aJO have similar projections. Most of the cells project from the AMMC to the ventral SEZ, and a smaller subset projects from the AMMC to the posterior SEZ (new data added as Figure 1—figure supplement 4). In contrast to previously identified zone C/E-projecting neurons, we found no evidence of aJO neurons that project only to the AMMC. Text was added to the first Results section to describe this experiment. Additionally, we added more discussion that considers the implications of this result (see conclusions below).

Conclusions for questions 1 and 3:

A) Activation of JO neurons targeted by previously described GAL4 lines that express in the different JO regions revealed that C/E-projecting neurons induce grooming, whereas A/B-projecting subtypes do not (new data added as Figure 1—figure supplement 3). Additionally, newly added co-stains show that aJO neurons project into zone C/E (new data added as Figure 1—figure supplement 3). Therefore, JO-induced antennal grooming is likely restricted to neurons projecting to zone C/E, which argues against the possibility raised by the reviewers that the grooming is caused by non-specific activation of any JO neurons.

B) We find evidence that at least two subsets of zone C/E JO neurons induce grooming. The previously described C/E neurons show no obvious projections to the SEZ, but instead terminate in the AMMC (41). In contrast most, if not all of the aJO neurons project into the AMMC and then to the SEZ, with none appearing to terminate in the AMMC (Figure 1—figure supplement 4). This indicates that the aJO projections belong to a previously undescribed group, leading us to conclude that two different populations of C/E-projecting neurons can elicit antennal grooming – one that projects only to the AMMC, and the other (aJO) that projects to the AMMC and SEZ. Future work will be required to determine how these two populations of JO sensory neurons interact with the antennal grooming circuitry.

C) The antennal displacement distance that induces grooming (31 +/- 6 μm) is within the range shown to induce calcium responses in previously described C/E JO neurons. Therefore, aJO neurons might also be activated by such large displacements, coupled with our finding that they are critical for the grooming response to these displacements.

Overview of major revisions:

A) We revised and extended the Results and Discussion to incorporate the new results and draw the conclusions mentioned above. Particularly, we added text to the Results section (last two paragraphs) called “A group of Johnston’s Organ mechanosensory neurons elicits antennal grooming” and Discussion section, “A distinct class of JO neurons detects antennal displacements and elicits grooming”.

B) Because there are likely two populations of zone C/E JO neurons that induce grooming, we no longer say aJO neurons are responsible for antennal grooming in the Abstract. Instead we state, “Mechanosensory chordotonal neurons detect displacements of the antennae and excite three different classes of functionally connected interneurons, which include two classes of brain interneurons and different parallel descending neurons.”

C) Given that additional zone C/E neurons induce grooming, and that aJO-LexA may target some of these non-aJO neurons (Figure 1—figure supplement 3, Figure 4—figure supplement 2), we now refer to this LexA line as targeting aJO+C/E neurons in the Results. We have revised our conclusions to more generally state that zone C/E neurons elicit antennal grooming, rather than just the aJO neurons. This required us to modify some text in the Results, Discussion, and labels in Figures 5, 6 and 7. The main part of the manuscript that addresses this is in the second paragraph of the Results section called “A putative circuit that elicits grooming in response to antennal displacement.”

*2) The Introduction is not broad enough. For example, the authors need to provide more background on CPGs, patterned motor behaviors, and grooming, and to place the behavior they are studying in the context of the larger body of work. For example, the authors only cite papers on stick insect walking when introducing stereotyped motor patterns – they need to invoke behaviors like song production, feeding, etc. and compare/contrast their behavior with other well-studied motor patterns. In addition, the authors also don't provide enough background on their previous study (*[69]*) – this study is in many ways a follow-up, but they expect the reader to have familiarity with this paper already*.

We now provide additional background information on stereotyped movement patterns, CPGs, the study of neurons that induce specific movements in *Drosophila*, and additional information on our previous work. However, on the latter point, our previous work was primarily focused on how grooming is organized into a sequence of movements, which is not what we intended this paper to be about. The only point that we want to introduce from this previous work is the finding that activating different subsets of neurons induces specific grooming movements. This sets up the identification of neurons that specifically elicit antennal grooming. We have added two additional sentences to the last paragraph of the Introduction that more completely describes this point from our previous work. Additionally, the reviewers asked us to compare and contrast grooming with other stereotyped motor patterns. We felt that this was better addressed in more detail in the Discussion, framed in the context of what we learned from the results presented in the paper. The last section of the Discussion discusses the antennal grooming circuit in the context of other command-like neurons that have been described in flies. We hope that the reviewers are satisfied with this approach.

*4) Functional connectivity versus synaptic connectivity: The data presented clearly show functional connectivity between neurons. However, there is only weak evidence (it is not ‘likely’), that functionally connected neurons are postsynaptic to each other since synaptic delays are known to be small relative to the temporal resolution of the GCaMP6s used. Their approach detects functional connectivity but not synaptic connectivity – which is fine. However, the authors should make this clear throughout the manuscript (and in the Abstract)*.

We made numerous changes throughout the manuscript to make clear that our evidence supports functional connectivity among the neurons. Where appropriate, we now use the terminology suggested by the reviewers, such as “neurons downstream” or “neurons upstream” (suggested in question 7). In other places we now refer to the neurons as “functionally connected” rather than connected.

*5) Resolution of the behavior: Given the authors' approach to scoring the behavior manually (and lumping all of antennal grooming into one category) it is not possible to examine whether activity of different neurons in the antennal grooming circuit leads to different types/aspects of behavior or different timing of the behavior or differences in trial-to-trial reproducibility. Given that the authors suggest in their Introduction and Discussion that this behavior is a good model for understanding the contribution of sensory drive and modulation to stereotyped motor programs, it would help to know how stereotyped the behavior is. The authors should add a paragraph to their Discussion of the future value of examining the grooming behavior of* Drosophila *at higher resolution (think Dragonfly or* Drosophila *flight/steering,* Drosophila *or bird song production, ant path navigation, or arm reaching). In other words, there is much more to get out of this system now that the circuit is mapped*.

We have added a paragraph to the Discussion that now addresses this important point. In brief, we discuss that the antennal grooming circuit might control parameters that our current analysis methods cannot detect. The level of behavioral analysis presented in this work does not allow for the detection of fine antennal grooming movements. Thus, higher resolution analysis of the kinematics of the leg movements may allow for testing the roles of the circuit in controlling different variables of antennal grooming. This text can be found in the last paragraph of the Discussion called “A command circuit that elicits variable grooming durations.”

*6) The differentiation of the classes of antennal grooming descending neurons (aDN) appears to be less specific compared to the two aLN-classes. For example, no information is given on aDN2 in*
Figure 3
*and its legend. In general throughout the text the number of DN-classes that are referred to is changing from two (in the subsection “Three different interneuron classes elicit antennal grooming”), to one (in the subsection “A putative circuit that elicits grooming in response to antennal displacement”) and to three (also in the aforementioned subsection). In order to justify the summary*
Figure 7*, it is important that the authors allow the reader to keep clear track for all data available on DN classes leading to this scheme*.

We made several revisions to the text to ensure that our logic for stating that there are three aDNs is consistent and clear throughout. First, we added text to the last sentence of the last paragraph of the Results section called “Three different interneuron classes elicit antennal grooming” that should help establish that there are three aDNs. This text reads: “However, we did not identify a spGAL4 combination that exclusively expresses in the third R18C11-targeted aDN (named aDN3)” Second, we added text to the Figure 2 legend that mentions aDN3. This text reads: “No spGAL4 combinations were identified that exclusively target aDN3.” Third, we revised text to the second sentence of the first paragraph of the Results section called “A putative circuit that elicits grooming in response to antennal displacement”, which now explicitly mentions all three aDNs. The sentence reads: “aBNs project to the AMMC and SEZ, following the aJO projections (Figure 3), whereas two aDNs (aDN1 and aDN2), and likely aDN3 project both to the SEZ and through the cervical connective to the ProNm (Figure 3, only aDN1 shown).” Fourth, to avoid confusion about why only aDN1 is shown in Figure 3, we added text to the legend stating: “aDN1 is shown as an example in (D), but there are additional aDNs (aDN2 and aDN3, see Figure 2 and Figure 2—figure supplement 2)”. Finally, we added text that further justifies the way the aDNs are depicted in the Figure 7 in the first paragraph of the Results section called “A circuit whose components elicit different durations of antennal grooming.” This text reads: “We propose that the aDNs can act in parallel because experiments to thermogenetically activate aDN1 or aDN2 alone indicate that they are each sufficient to induce antennal grooming (Figure 2).”

*7) In the third paragraph of the subsection “Connectivity among the antennal grooming neuronal classes”, based on the neuroanatomical analysis of the branching patterns of the described sensory and brain interneurons, the authors conclude specific connectivity between them. I agree that their data show that these neurons are likely to belong to the same circuit, but I do not see proof of an explicit synaptic connectivity between them: (i) the anatomical analysis of the branching patterns has not the scale to allow for that – a scale of 25µm as given in the figures is not sufficient to resolve connectivity (e.g. see*
Figure 5*) and (ii) the same holds for the calcium transients recorded (e.g. see*
Figure 6*). Such explicit interpretation is not justified. A terminology like “neurons downstream” or “neurons upstream” is appropriate*.

See question 4 for the revisions that we made to the manuscript. Of note, we incorporated the reviewers’ suggestion to use terminology, such as neurons upstream or downstream.

*8) With regards to the subsection “A command circuit that elicits variable grooming durations”, all neural circuits are composed of “multiple neurons”. It is not clear which point the authors want to make*.

We agree with the reviewers that we did not sufficiently describe the significance of circuits that consist of multiple neurons. We clarified this point in the third paragraph of the subsection “A command circuit that elicits variable grooming durations” (in the Discussion). We describe how the antennal grooming circuit and other well-described circuits are organized such that some neurons induce movements that persist beyond their initial activation, whereas other neurons must be continually activated to continue the movement. We point out that this type of organization appears to be common among sensory-connected neuronal circuits that induce specific movements.

*Specifically, information is lacking on how the new findings relate to previous findings on neurons that serve command-like functions in insect behavior (work on courtship behavior in locusts and crickets; mostly Hedwig and coauthors) and in the fruit fly? With respect to the fruit fly I was missing a discussion on what the new study adds to the previous work on command neurons in the control of fruit fly courtship behavior (e.g.*
[78]*), aversive motor behavior (*[19]*) and walking direction (*[2]*)?*

We added a new section to the Discussion entitled “Neurons that command specific movements in insects.” This section describes how work in flies and other insects suggests that the use of dedicated neurons to command specific movements is a common mechanism of behavioral control in insects. We then insert the caveat that these command-like neurons may be embedded in larger, more dynamic neural networks in which neurons that elicit one movement also participate in other movements. We cite work from leech that supports this possibility.

*9) Is there really a reciprocal excitation loop in this circuit, and is it responsible for persistent grooming behavior evoked by optogenetic stimulation? The authors place a great deal of emphasis on the finding of an apparent reciprocal excitation loop between the aLN1 and aLN2 local interneurons (*Figure 7*), and invoke it to explain their interesting observation that activation of these interneurons can produce persistent grooming (*Figure 7*). Unfortunately, due to limitations in the specificity of the reagents used to perform these experiments, the evidence of reciprocal excitation is unconvincing and it is very difficult to correlate the physiological (imaging) and behavioral data*.

Questions 9, 10, and 11 all relate to the strength of evidence supporting aBN1/aBN2 reciprocal functional connections. We address the specific points for each of these questions and then summarize the revisions related to all of them at the end.

*Firstly, the evidence of reciprocal connectivity is weak. The degree of anatomical overlap between aLN1 and aLN2 neurons is very limited (*Figure 5
*– by the way, please indicate with dashed outlines the location of the AMMC, posterior and ventral SEZ on these figures for the readers' benefit). Although the evidence for functional connectivity from LN1→LN2 neurons is convincing (*Figure 6*), the evidence for the reciprocal connection (*Figure 6*) is not. Apparently (based on the information provided in*
Figure 6—figure supplement 3*), to test for the latter connection the authors used an LN2-LexA driver to express CsChrimson in these neurons (in contrast to the reverse experiment where they used an LN1-GAL4 driver to express the opsin). This LN2-LexA driver is extremely broadly expressed, in the central brain and optic lobes (*Figure 4—figure supplement 1*). Without specific 2-photon targeted activation of the LN2 neurons themselves, it is difficult to be confident that the excitatory drive on the LN1 neurons in*
Figure 6
*is indeed due to input from the aLN2 cells*.

The reviewers requested that we used dashed outlines to mark the different projection regions. We realized with this comment that there was confusion about the meaning of AMMC, posterior and ventral SEZ in this context. We did not intend for these to refer to distinct regions of the neuropile, but to the three major regions defined by the aJO projections shown in Figure 3 (colored arrows). Therefore, defining specific regions using dashed outlines would be arbitrary and misleading. We have tried to make this clearer in Figure 5 by adding the same colored arrows shown in Figure 3 to identify each projection region containing overlap between the different neuronal pairs. We added text to the Figure 5 legend that describes what the arrows mean. This text reads: “Overlap between different projections of the LexA and spGAL4-targeted neurons is indicated by different colored arrows: (A-C, E, F) AMMC projections white arrows, (A) posterior SEZ projection (yellow arrow), (B-H) ventral SEZ projections (red arrows).”

We agree that the aBN2-LexA driver expresses in additional neurons, raising the possibility that aBN2 is not the upstream cause of the aBN1 increased calcium response shown in Figure 6. Therefore we added text in the Results that describes this caveat. This text is in the fourth paragraph of the Results section called “Functional connectivity among the antennal grooming neuronal classes”, and states: “Further, because the aBN2-LexA driver used to express CsChrimson in aBN2 also expresses in neurons in other parts of the brain (Figure 4—figure supplement 1), we cannot rule out these other neurons as causing the aBN1 calcium response.”

*10) Furthermore, the magnitude of the average LN1 response to “LN2 stimulation” in*
Figure 6
*is much weaker than that of the LN1 → LN2 stimulation in*
Figure 6*. Close inspection of the supplemental data indicates that even this weak average recording is misleading, because only a fraction of the flies tested, and only a small fraction of trials, yielded any response at all (*Figure 6—figure supplement 3*, LN2 to LN1). This is very different from a result in which the neurons showed a weak response in most of the flies in most of the trials. This is all the more puzzling given that there are more aLN2 than aLN1 cells (*Figure 7*), and that the activation of neurons in the aLN2-spGAL4-2 line has the strongest effect to promote grooming of any of the lines characterized in the paper (*Figures 2 and 7*). Thus, the evidence for an LN2→LN1 connection that is relevant to grooming is, in my opinion, weak and unconvincing*.

We revised the manuscript to more completely describe the data that contributes to the average traces shown in Figure 6. Text was added to the second paragraph of the “GCaMP6s imaging and analysis of calcium responses” section of the Materials and methods. This more explicitly describes which imaging trials were used for data shown in Figure 6. We also labeled the trials from the raw imaging data shown in Figure 6—figure supplement 2 and Figure 6—figure supplement 3 that contribute to the average traces shown in Figure 6.

*At the very least, the circuit diagram in*
Figure 7
*should be re-drawn to show that the connection from aLN2→aLN1 is far weaker and more questionable than the reverse connection (perhaps using a dashed line and question mark), and this issue and associated caveats should be clearly spelled out in the Discussion.*

Figure 7 is revised to weaken our statement about a functional connection between aBN2 and aBN1. We redrew the connection from aBN2 to aBN1 using a dotted gray line to emphasize that the connection is not as well supported as the other connections in the diagram. Text was added to the figure legend that describes the meaning of the gray arrow. We also added text to the Figure 7 Results section to explain the aBN2 to aBN1 connection caveats (discussed more below).

*11) In addition, if the authors really want to make the case that this putative reciprocal excitatory loop is responsible for the persistent grooming behavior they observe, then they should perform an epistasis experiment in which they demonstrate that persistent grooming triggered by aLN2 activation (but perhaps not grooming per se) is blocked by inactivating aLN1 cells, and/or vice-versa*.

The manuscript is now revised to deemphasize the putative reciprocal excitation between aBN1 and aBN2. We describe the caveats associated with the experiment to test the aBN2 to aBN1 functional connectivity. The major revisions are described below.

Description of major revisions for questions 9-11:

A) We removed mention of the reciprocal excitation from the Abstract.

B) We added description of the caveats associated with the aBN2 to aBN1 functional connectivity experiment, and explicitly state that the connection is only weakly supported by the data. This description is found in the fourth paragraph of the Results section called “Functional connectivity among the antennal grooming neuronal classes.” As the reviewers pointed out, we now indicate that only high-intensity red light activation of aBN2 could cause a weak calcium response in aBN1. We mention that this response was inconsistent, as only three out of five flies tested showed increases in calcium in aBN1. We further point out that the aBN2-LexA driver also expresses in neurons in other parts of the brain (discussed above).

C) We removed most of the discussion of reciprocal excitation as an underlying mechanism causing the persistence. However, reciprocal excitation is briefly mentioned in the sixth paragraph of the Discussion section called “A command circuit that elicits variable grooming durations” as one of two possible mechanisms that could cause the persistence. This text reads: “Although our work only provides weak evidence for reciprocal excitation between the aBNs, a mechanism whereby reciprocal connections maintain their excitation offers a plausible explanation for the persistence that we observe. Alternatively, aBN1 and aBN2 could have intrinsic membrane properties that allow them to produce prolonged responses to brief excitation. Thus, future experiments will examine whether reciprocal excitation or intrinsic membrane properties of the aBNs produce persistent grooming (50).”

*Finally, the authors should point out that in their antennal displacement assay, only a small fraction of stimulation trials elicited persistent grooming, and that even in those cases the persistence appears to be in the grooming of the eyes/head, not the antenna (*Figure 4—figure supplement 3*). So the biological relevance of this persistence phenomenon, interesting though it may be, is not obvious*.

The fourth paragraph of the Discussion section entitled “Biological functions of the antennal grooming circuit” addresses the possible biological relevance of persistent grooming. We discuss how our results do not identify a natural stimulus that induces reliable persistent grooming. However, persistence was previously described in the grooming responses of other animals, where it was shown to be at least partially dependent on the nature of the stimulus (e.g. stimulus strength, duration, or frequency). Thus, future work will explore the possibility that specific stimulus parameters can evoke persistent antennal grooming in flies.